# Spectral Learning of Shared Dynamics Between Generalized-Linear Processes

**Lucine L. Oganesian**
Ming Hsieh Department of Electrical and Computer Engineering
University of Southern California
Los Angeles, CA
`loganesi@usc.edu`

**Omid G. Sani**
Ming Hsieh Department of Electrical and Computer Engineering
University of Southern California
Los Angeles, CA
`omid.ghasemsani@usc.edu`

**Maryam M. Shanechi** [*]
Ming Hsieh Department of Electrical and Computer Engineering
Thomas Lord Department of Computer Science
Alfred E. Mann Department of Biomedical Engineering
Neuroscience Graduate Program
University of Southern California
Los Angeles, CA
`shanechi@usc.edu`

## Abstract

Generalized-linear dynamical models (GLDMs) remain a widely-used framework within neuroscience for modeling time-series data, such as neural spiking activity or categorical decision outcomes. Whereas the standard usage of GLDMs is to model a single data source, certain applications require jointly modeling two generalized-linear time-series sources while also dissociating their shared and private dynamics. Most existing GLDM variants and their associated learning algorithms do not support this capability. Here we address this challenge by developing a multi-step analytical subspace identification algorithm for learning a GLDM that explicitly models shared vs. private dynamics within two generalized-linear time-series. In simulations, we demonstrate our algorithm's ability to dissociate and model the dynamics within two time-series sources while being agnostic to their respective observation distributions. In neural data, we consider two specific applications of our algorithm for modeling discrete population spiking activity with respect to a secondary time-series. In both synthetic and real data, GLDMs learned with our algorithm more accurately decoded one time-series from the other using lower-dimensional latent states, as compared to models identified using existing GLDM learning algorithms.

---

[*]Corresponding author. Project code: https://github.com/ShanechiLab/PGLDM

38th Conference on Neural Information Processing Systems (NeurIPS 2024).

# 1  Introduction

Generalized-linear dynamical models (GLDMs) are a commonly used framework for modeling dynamics using a low-dimensional latent variable that evolves over time [1–3]. Due to their interpretability, data efficiency, and amenability to real-time engineering operations, GLDMs remain a widely popular tool in neuroscience for modeling time-series data, whether it be Poisson spiking neural activity, Bernoulli/Binomial categorical task variables, or Gaussian behavior [4–9]. Whereas most existing GLDM variants and their associated analytical learning algorithms focus on modeling the dynamics within a single time-series, there exist applications that require explicit dissociation of shared vs. private dynamics within two generalized-linear observation sources. For example, such functionality is helpful when modeling the dynamical relationship between recorded neural activity and certain behaviors of interest [10–15].

Here we fill these methodological gaps by deriving a novel covariance-based subspace system identification (SSID) algorithm that is capable, with its multi-staged learning approach, of identifying the shared dynamics between two generalized-linear time-series with priority, before modeling the dynamics private to each observation. We design the method to seamlessly generalize to different observation distributions, such as Poisson or Bernoulli. To illustrate the method, we first show in simulations that our method successfully dissociates the shared dynamics within two generalized-linear time-series, agnostic of their respective observation models; to compare against existing GLDM methods, we focused on Poisson, Bernoulli, and Gaussian generalized-linear observations. Next we demonstrate our method on two public non-human primate (NHP) datasets of discrete population spiking activity recorded from different brain regions and during different contexts [16–18]. Compared with existing Poisson GLDMs and their learning algorithms, our method learned models that more accurately decoded one time-series from the other using lower-dimensional latent states, suggesting improved learning of shared dynamics.

# 2  Background

For linear state-space models with continuous Gaussian observations, subspace system identification (SSID) theory provides computationally efficient non-iterative algorithms for analytically learning state-space models, both with and without identification of shared dynamics [14, 19–24]. These methods, however, either are not applicable to generalized-linear time-series with non-Gaussian observations [14, 19, 20, 23] or do not have the ability to dissociate shared vs. private dynamics between two time-series [21, 22, 24]. To help with the exposition of our method in section 3, we first review standard covariance-based SSID and an existing SSID method for modeling Poisson point-processes [21], a widely-used class of generalized-linear observations.

## 2.1  Standard covariance-based SSID

The standard formulation for a linear state-space model with continuous Gaussian observations is as

$$\begin{cases} \mathbf{x}_{k+1} & = & \boldsymbol{A}\mathbf{x}_k + \mathbf{w}_k \\ \mathbf{r}_k & = & \boldsymbol{C}_\mathbf{r}\mathbf{x}_k + \mathbf{v}_k \end{cases} \tag{1}$$

where $\mathbf{x}_k \in \mathbb{R}^{n_x}$ is the latent state variable, $\mathbf{r}_k \in \mathbb{R}^{n_r}$ corresponds to continuous Gaussian observations, and $\mathbf{w}$ and $\mathbf{v}$ are state and observation noise terms, respectively, with distributions $\mathcal{N}(\mathbf{w}_k; \mathbf{0}, \boldsymbol{Q})$ and $\mathcal{N}(\mathbf{v}_k; \mathbf{0}, \boldsymbol{R})$, and cross-covariance $\boldsymbol{S}$. Further, we define $\boldsymbol{G} := \mathrm{Cov}(\mathbf{x}_{k+1}, \mathbf{r}_k)$, as the covariance between future latent state and current observation, and $\boldsymbol{\Lambda}_{\mathbf{r}_0} := \mathrm{Cov}(\mathbf{r}_k, \mathbf{r}_k)$, as the instantaneous covariance of the observations. Standard covariance-based SSID learns the parameters of a latent dynamical system $\Theta = (\boldsymbol{A}, \boldsymbol{C}_\mathbf{r}, \boldsymbol{G}, \boldsymbol{\Lambda}_{\mathbf{r}_0})$ given training samples $\boldsymbol{r}_k$ and hyperparameter $n_x$ that specifies the latent state dimensionality. To do so, a future-past Hankel matrix, $\boldsymbol{H}_\mathbf{r}$, is first constructed from the cross-covariances of the system's linear observations as [19, 20]

$$\boldsymbol{H}_\mathbf{r} := \mathrm{Cov}(\mathbf{r}_f, \mathbf{r}_p) = \begin{bmatrix} \boldsymbol{\Lambda}_{\mathbf{r}_i} & \boldsymbol{\Lambda}_{\mathbf{r}_{i-1}} & \cdots & \boldsymbol{\Lambda}_{\mathbf{r}_1} \\ \boldsymbol{\Lambda}_{\mathbf{r}_{i+1}} & \boldsymbol{\Lambda}_{\mathbf{r}_i} & \cdots & \boldsymbol{\Lambda}_{\mathbf{r}_2} \\ \vdots & \vdots & \cdots & \vdots \\ \boldsymbol{\Lambda}_{\mathbf{r}_{2i-1}} & \boldsymbol{\Lambda}_{\mathbf{r}_{2i-2}} & \cdots & \boldsymbol{\Lambda}_{\mathbf{r}_i} \end{bmatrix}, \quad \mathbf{r}_f := \begin{bmatrix} \mathbf{r}_i \\ \vdots \\ \mathbf{r}_{2i-1} \end{bmatrix}, \quad \mathbf{r}_p := \begin{bmatrix} \mathbf{r}_0 \\ \vdots \\ \mathbf{r}_{i-1} \end{bmatrix}, \tag{2}$$

where the integer $i$ denotes the user-specified maximum temporal lag (i.e., horizon) used to construct $\boldsymbol{H}_\mathbf{r}$ and $\boldsymbol{\Lambda}_{\mathbf{r}_\tau} := \mathrm{Cov}(\mathbf{r}_{k+\tau}, \mathbf{r}_k)$ is the $\tau$-th lag cross-covariance for any timepoint $k$, under time-stationary assumptions. We note that the rank of $\boldsymbol{H}_\mathbf{r}$ must be at least $n_x$ in order to identify a

model with a latent dimension of $n_x$. Thus the user-specified horizon $i$ must satisfy $i \times n_r \geq n_x$. Covariance-based SSID then decomposes $\boldsymbol{H_r}$ into a product of observability ($\boldsymbol{\Gamma_r}$) and controllability ($\boldsymbol{\Delta}$) matrices as [19, 20]

$$\boldsymbol{H_r} \stackrel{\text{SVD}}{=} \boldsymbol{\Gamma_r \Delta} = \begin{bmatrix} \boldsymbol{C_r} \\ \boldsymbol{C_r A} \\ \vdots \\ \boldsymbol{C_r A}^{i-1} \end{bmatrix} \begin{bmatrix} \boldsymbol{A}^{i-1}\boldsymbol{G} & \cdots & \boldsymbol{AG} & \boldsymbol{G} \end{bmatrix} \tag{3}$$

where $\boldsymbol{G}$ is defined as above. The factorization of $\boldsymbol{H_r}$ is done by computing a singular value decomposition (SVD) of $\boldsymbol{H_r}$ and keeping the top $n_x$ singular values and corresponding singular vectors. From the factors of $\boldsymbol{H_r}$, $\boldsymbol{C_r}$ is read off as the first $n_r$ rows of $\boldsymbol{\Gamma_r}$ and $\boldsymbol{G}$ is read off as the last $n_r$ columns of $\boldsymbol{\Delta}$. $\boldsymbol{A}$ is learned by solving $\overline{\boldsymbol{\Gamma}}_r = \underline{\boldsymbol{\Gamma}}_r \boldsymbol{A}$, where $\overline{\boldsymbol{\Gamma}}_r$ and $\underline{\boldsymbol{\Gamma}}_r$ denote $\boldsymbol{\Gamma_r}$ from which the top or bottom $n_r$ rows have been removed, respectively. This optimization problem has the following closed-form least-squares solution $\boldsymbol{A} = \underline{\boldsymbol{\Gamma}}_r^\dagger \overline{\boldsymbol{\Gamma}}_r$, with $\dagger$ denoting the pseudo-inverse operation. The final parameter $\boldsymbol{\Lambda}_{\boldsymbol{r}_0}$ is computed as the empirical covariance of $\boldsymbol{r}_k$. See appendix A.4 on how $(\boldsymbol{G}, \boldsymbol{\Lambda}_{\boldsymbol{r}_0})$ specify $(\boldsymbol{Q}, \boldsymbol{R}, \boldsymbol{S})$.

## 2.2 SSID for a single generalized-linear time-series

There has been some work extending SSID to generalized-linear time-series, such as Poisson and Bernoulli observations [21, 24]. These methods, however, only learn the dynamics of a single generalized-linear time-series rather than model shared vs. private dynamics between two time-series. Here we present one of our baselines, PLDSID [21], which models a single Poisson time-series, as an example. A Poisson linear dynamical system (PLDS) model is defined as

$$\begin{cases} \mathbf{x}_{k+1} & = & \boldsymbol{A}\mathbf{x}_k + \mathbf{w}_k \\ \mathbf{r}_k & = & \boldsymbol{C_r}\mathbf{x}_k + \boldsymbol{b} \\ \mathbf{y}_k \mid \mathbf{r}_k & \sim & \text{Poisson}(\exp(\mathbf{r}_k)) \end{cases} \tag{4}$$

where $\mathbf{x}_k \in \mathbb{R}^{n_x}$ is the latent state as before and $\mathbf{y}_k \in \mathbb{R}^{n_y}$ corresponds to discrete (e.g., neural spiking) observations which, conditioned on the latent process $\mathbf{r}_k$, are Poisson-distributed with a rate equal to the exponential of $\mathbf{r}_k$ (i.e., log-rate). Finally, $\mathbf{w}_k$ is Gaussian-distributed state noise with covariance parameter $\boldsymbol{Q}$, as before, and $\boldsymbol{b}$ is a constant baseline log-rate. The PLDS model is commonly used for modeling Poisson process events, such as neural spiking activity [2, 4, 6, 21, 25]. Buesing et al. [21] developed a SSID algorithm, termed PLDSID, to learn the PLDS model parameters $\Theta_{\text{PLDS}} = (\boldsymbol{A}, \boldsymbol{C_r}, \boldsymbol{b}, \boldsymbol{Q})$ given training samples $\boldsymbol{y}_k$ and hyperparameter $n_x$.

Standard covariance-based SSID algorithms (section 2.1) are not directly applicable to Poisson-distributed observations. This is because the log-rates $\mathbf{r}_k$ that are linearly related to the latent states in equation (4) are not observable in practice – rather, only a stochastic Poisson emission from them (i.e., $\mathbf{y}_k$) is observed. As a result, the second moments constituting $\boldsymbol{H_r}$ (i.e., $\boldsymbol{\Lambda}_{\boldsymbol{r}_\tau}$) cannot be directly estimated. The critical insight by Buesing et al. [21] was to leverage the log link function (i.e., $\exp^{-1}$) and the known conditional distribution $\mathbf{y}_k|\mathbf{r}_k$ to compute the first ($\boldsymbol{\mu}_{\mathbf{r}^\pm}$) and second ($\boldsymbol{\Lambda}_{\mathbf{r}^\pm}$) moments of the log-rate $\mathbf{r}_k$ from the first ($\boldsymbol{\mu}_{\mathbf{y}^\pm}$) and second ($\boldsymbol{\Lambda}_{\mathbf{y}^\pm}$) moments of the discrete observations $\mathbf{y}_k$. The $\pm$ denotes that moments are computed for the future-past stacked vector of observations $\mathbf{r}^\pm := \begin{bmatrix} \mathbf{r}_f^T & \mathbf{r}_p^T \end{bmatrix}^T$ and $\mathbf{y}^\pm := \begin{bmatrix} \mathbf{y}_f^T & \mathbf{y}_p^T \end{bmatrix}^T$, where

$$\boldsymbol{\mu}_{\mathbf{r}^\pm} := E[\mathbf{r}^\pm] \quad \boldsymbol{\mu}_{\mathbf{y}^\pm} := E[\mathbf{y}^\pm] \quad \boldsymbol{\Lambda}_{\mathbf{r}^\pm} := \text{Cov}(\mathbf{r}^\pm, \mathbf{r}^\pm) \quad \boldsymbol{\Lambda}_{\mathbf{y}^\pm} := \text{Cov}(\mathbf{y}^\pm, \mathbf{y}^\pm).$$

To compute moments of the log-rate, Buesing et al. [21] derived the following moment conversion

$$\begin{array}{rcl} \boldsymbol{\mu}_{\mathbf{r}_m^\pm} & = & 2\ln(\boldsymbol{\mu}_{\mathbf{y}_m^\pm}) - \dfrac{1}{2}\ln(\boldsymbol{\Lambda}_{\mathbf{y}_{mm}^\pm} + \boldsymbol{\mu}_{\mathbf{y}_m^\pm}^2 - \boldsymbol{\mu}_{\mathbf{y}_m^\pm}) \\ \boldsymbol{\Lambda}_{\mathbf{r}_{mm}^\pm} & = & \ln(\boldsymbol{\Lambda}_{\mathbf{y}_{mm}^\pm} + \boldsymbol{\mu}_{\mathbf{y}_m^\pm}^2 - \boldsymbol{\mu}_{\mathbf{y}_m^\pm}) - \ln(\boldsymbol{\mu}_{\mathbf{y}_m^\pm}^2) \\ \boldsymbol{\Lambda}_{\mathbf{r}_{mn}^\pm} & = & \ln(\boldsymbol{\Lambda}_{\mathbf{y}_{mn}^\pm} + \boldsymbol{\mu}_{\mathbf{y}_m^\pm}\boldsymbol{\mu}_{\mathbf{y}_n^\pm}) - \ln(\boldsymbol{\mu}_{\mathbf{y}_m^\pm}\boldsymbol{\mu}_{\mathbf{y}_n^\pm}) \end{array} \tag{5}$$

where $m \neq n$ correspond to different indices of the first and second moments of the future-past stacked observation vectors $\mathbf{r}^\pm$ and $\mathbf{y}^\pm$, and $n, m = 1, \cdots, Kn_y$ where $K$ is the total number of time points. With the first and second moments computed in the moment conversion above, the baseline log rate $\boldsymbol{b}$ parameter is read off the first $n_r$ rows of $\boldsymbol{\mu}_{\mathbf{r}^\pm}$ and the Hankel matrix, $\boldsymbol{H_r}$, is constructed as per equation (2). From here, we can proceed with the standard covariance-based SSID algorithm using $\boldsymbol{H_r}$, as outlined in section 2.1. Discussion regarding learning the state noise covariance parameters (e.g., $\boldsymbol{Q}$) is postponed to appendix section A.3, where we use an approach that – unlike Buesing et al. [21] – ensures validity of learned noise statistics.

## 3 Method

### 3.1 Model definition and assumptions

Both the standard linear state-space model (equation (1)) and PLDS model (equation (4)) only model a single observation on its own. To enable identification of shared and private dynamics between two generalized-linear time-series, we write the following general multi-observation GLDM

$$
\begin{cases}
\mathbf{x}_{k+1} & = & \boldsymbol{A}\mathbf{x}_k + \mathbf{w}_k \\
\mathbf{r}_k & = & \boldsymbol{C}_{\mathbf{r}}\mathbf{x}_k + \mathbf{v}_k + \boldsymbol{b} \\
\mathbf{z}_k & = & \boldsymbol{C}_{\mathbf{z}}\mathbf{x}_k + \boldsymbol{\epsilon}_k + \boldsymbol{d} \\
\mathbf{y}_k|\mathbf{r}_k & \sim & \mathcal{P}_{\mathbf{y}|\mathbf{r}}(\mathbf{y}_k;\ g(\mathbf{r}_k)) \\
\mathbf{t}_k|\mathbf{z}_k & \sim & \mathcal{P}_{\mathbf{t}|\mathbf{z}}(\mathbf{t}_k;\ h(\mathbf{z}_k))
\end{cases}
\tag{6}
$$

where $\mathbf{x}_k$, $\mathbf{r}_k$, $\mathbf{w}_k$, $\mathbf{v}_k$, and $\boldsymbol{b}$ are defined as in equations (1) and (4). We introduce $\mathbf{t}_k \in \mathbb{R}^{n_t}$ to represent the second generalized-linear observation time-series, $\mathbf{z}_k \in \mathbb{R}^{n_z}$ (with $n_t = n_z$ by construction) to represent the latent process underlying this second observation, $\boldsymbol{\epsilon}_k \sim \mathcal{N}(\boldsymbol{\epsilon}_k; \mathbf{0}, \boldsymbol{F})$ to represent the associated noise term, and $\boldsymbol{d}$ to represent the associated baseline value. We generically denote the probability distribution for $\mathbf{y}_k$ conditioned on the latent $\mathbf{r}_k$ with $\mathcal{P}_{\mathbf{y}}$. For example, in the PLDS model $\mathcal{P}_{\mathbf{y}|\mathbf{r}} := \mathrm{Poisson}(\exp(\mathbf{r}_k))$. $\mathcal{P}_{\mathbf{t}|\mathbf{z}}$ is defined similarly but for $\mathbf{t}_k$ and $\mathbf{z}_k$. Finally, $g(\cdot)$ and $h(\cdot)$ correspond to the link function in the generalized-linear model, for example $g(\mathbf{r}_k) = \exp(\mathbf{r}_k)$ in PLDS models. In order to dissociate between shared and private dynamics within the observation time-series, we introduce the following definition:

**Definition 3.1.** We take the system to be written in a block structure form as defined below [14], allowing us to dissociate shared from private latents

$$
\boldsymbol{A} = \begin{bmatrix} \boldsymbol{A}_{11} & \mathbf{0} & \mathbf{0} \\ \boldsymbol{A}_{21} & \boldsymbol{A}_{22} & \mathbf{0} \\ \mathbf{0} & \mathbf{0} & \boldsymbol{A}_{33} \end{bmatrix} \quad \boldsymbol{C}_{\mathbf{z}} = \begin{bmatrix} \boldsymbol{C}_{\mathbf{z}}^{(1)} & \mathbf{0} & \boldsymbol{C}_{\mathbf{z}}^{(3)} \end{bmatrix} \quad \boldsymbol{C}_{\mathbf{r}} = \begin{bmatrix} \boldsymbol{C}_{\mathbf{r}}^{(1)} & \boldsymbol{C}_{\mathbf{r}}^{(2)} & \mathbf{0} \end{bmatrix} \quad \mathbf{x} = \begin{bmatrix} \mathbf{x}^{(1)} \\ \mathbf{x}^{(2)} \\ \mathbf{x}^{(3)} \end{bmatrix} \tag{7}
$$

where $\mathbf{x}_k^{(1)} \in \mathbb{R}^{n_1}$ corresponds to latent states that drive both $\mathbf{z}_k$ and $\mathbf{r}_k$, $\mathbf{x}_k^{(2)} \in \mathbb{R}^{n_2}$ corresponds to states that only drive $\mathbf{r}_k$, and $\mathbf{x}_k^{(3)} \in \mathbb{R}^{n_3}$ corresponds to states that only drive $\mathbf{z}_k$ – with total states $n_x = n_1 + n_2 + n_3$. The parameter $\boldsymbol{G}$ can also be written in block partition format such that

$$
\boldsymbol{G} = E\left[\begin{bmatrix} \mathbf{x}_{k+1}^{(1)} \\ \mathbf{x}_{k+1}^{(2)} \\ \mathbf{x}_{k+1}^{(3)} \end{bmatrix} \mathbf{r}_k^T\right] - E\left[\begin{bmatrix} \mathbf{x}_{k+1}^{(1)} \\ \mathbf{x}_{k+1}^{(2)} \\ \mathbf{x}_{k+1}^{(3)} \end{bmatrix}\right] E[\mathbf{r}_k]^T = \begin{bmatrix} E[\mathbf{x}_{k+1}^{(1)}\mathbf{r}_k^T] \\ E[\mathbf{x}_{k+1}^{(2)}\mathbf{r}_k^T] \\ E[\mathbf{x}_{k+1}^{(3)}\mathbf{r}_k^T] \end{bmatrix} - \begin{bmatrix} E[\mathbf{x}_{k+1}^{(1)}]E[\mathbf{r}_k]^T \\ E[\mathbf{x}_{k+1}^{(2)}]E[\mathbf{r}_k]^T \\ E[\mathbf{x}_{k+1}^{(3)}]E[\mathbf{r}_k]^T \end{bmatrix} = \begin{bmatrix} \boldsymbol{G}^{(1)} \\ \boldsymbol{G}^{(2)} \\ \boldsymbol{G}^{(3)} \end{bmatrix}.
$$

We can further simplify the definition of $\boldsymbol{G}$ using the following assumptions:

**Assumption 3.2.** *The state noise covariance $\boldsymbol{Q}$ is assumed to have a block diagonal structure such that $\boldsymbol{Q} = \mathrm{diag}(\boldsymbol{Q}^{(1,2)}, \boldsymbol{Q}^{(3)})$, where $\boldsymbol{Q}^{(1,2)}$ is a square matrix of dimension $n_1 + n_2$ and $\boldsymbol{Q}^{(3)}$ is a square matrix of dimension $n_3$. Formally, the superscript notation $(1, 2)$ designates attribution of the parameter to the first and second set of latent states.*

**Assumption 3.3.** *Initial latent states are assumed to be mutually-independent, making $\mathrm{Cov}(\mathbf{x}_0, \mathbf{x}_0)$ diagonal.*

These assumptions allow us to fully decouple the private latent states of the secondary time-series ($\mathbf{x}^{(3)}$) from the latent states driving the primary time-series ($\mathbf{x}^{(1)}, \mathbf{x}^{(2)}$). As a result, we can take $\boldsymbol{G}^{(3)} = \mathrm{Cov}(\mathbf{x}_{k+1}^{(3)}, \mathbf{r}_k) = \mathbf{0}$. From the perspective of state estimation, this simplification implies that $\mathbf{r}_k$ provides no information to help estimate $\mathbf{x}_{k+1}^{(3)}$, for all $k$; this understanding is consistent with our definition of shared and private states.

### 3.2 Prioritized generalized-linear dynamical modeling (PGLDM)

Our method, which we term Prioritized Generalized-Linear Dynamical Modeling (PGLDM), uses a multi-staged learning approach to model a primary generalized-linear time-series while prioritizing identification of the dynamics shared with a secondary time-series. Note, "primary" refers to the data source whose modeling is of primary interest and that can optionally be used to predict the

secondary data source. For example, within the context of decoding continuous behaviors from discrete population spiking activity, Poisson observations (i.e., neural activity) are the primary time-series whereas Gaussian observations (e.g., kinematics) are the secondary time-series. During stage 1, shared dynamics are learned using both observations. In stage 2, any private dynamics in the primary time-series are optionally learned. This two-staged approach allows prioritized learning of shared dynamics in the sense that latent states will be dedicated to explaining non-shared dynamics in the primary time-series only if there are enough latent states to explain the shared dynamics. Finally, an optional stage 3 allows identification of the dynamics private to the secondary time-series.

Below we will outline the first two stages of our algorithm and leave the optional stage 3 to appendices A.1.4 and A.9. A crucial component of our method is a new covariance-based SSID algorithm for identifying shared and private dynamics between two observation time-series using their first and second moments only. This covariance-based algorithm is what enables our method to be applicable to generalized-linear observations, which we will expand on further in section 3.2.3. We first present this new covariance-based SSID algorithm, or equivalently PGLDM for linear state-space models with continuous Gaussian observations (the first three lines of equation (6)), before showing support for generalized-linear time-series broadly. The derivation of PGLDM is provided in appendix A.1.

### 3.2.1 Stage 1: shared dynamics

In the first stage, our algorithm identifies the parameter set corresponding to the shared dynamical subspace, $(\boldsymbol{A}_{11}, \boldsymbol{C}_{\mathbf{r}}^{(1)}, \boldsymbol{C}_{\mathbf{z}}^{(1)}, \boldsymbol{b}, \boldsymbol{d})$, given hyperparameter $n_1$ and using both $\mathbf{z}_k$ and $\mathbf{r}_k$, both of which are observable in the Gaussian case (i.e., equation (1) or equivalently the first three lines of equation (6)). To do this, we first construct a Hankel matrix between future observations of the secondary process and past observations of the primary process

$$\boldsymbol{H}_{\mathbf{zr}} := \mathrm{Cov}(\mathbf{z}_f, \mathbf{r}_p) = \begin{bmatrix} \boldsymbol{\Lambda}_{\mathbf{zr}_i} & \boldsymbol{\Lambda}_{\mathbf{zr}_{i-1}} & \cdots & \boldsymbol{\Lambda}_{\mathbf{zr}_1} \\ \boldsymbol{\Lambda}_{\mathbf{zr}_{i+1}} & \boldsymbol{\Lambda}_{\mathbf{zr}_i} & \cdots & \boldsymbol{\Lambda}_{\mathbf{zr}_2} \\ \vdots & \vdots & \cdots & \vdots \\ \boldsymbol{\Lambda}_{\mathbf{zr}_{2i-1}} & \boldsymbol{\Lambda}_{\mathbf{zr}_{2i-2}} & \cdots & \boldsymbol{\Lambda}_{\mathbf{zr}_i} \end{bmatrix}, \tag{8}$$

with $\mathbf{z}_f := \begin{bmatrix} \mathbf{z}_i & \ldots & \mathbf{z}_{2i-1} \end{bmatrix}^T$ and $\mathbf{r}_p$ defined as in equation (2). Although equation (8) uses the same horizon for both observations, in practice we implement the method for a more general version with distinct horizon values $i_r$ for the primary observations and $i_z$ for the secondary observations, resulting in $\boldsymbol{H}_{\mathbf{zr}} \in \mathbb{R}^{i_z * n_z \times i_r * n_r}$. This allows users to independently specify the horizons for the two observations, which can improve modeling accuracy especially if the two observations have very different dimensionalities (see section 4.2 and appendix A.1.6). After constructing $\boldsymbol{H}_{\mathbf{zr}}$, we decompose it using SVD and keep the top $n_1$ singular values and their corresponding singular vectors

$$\boldsymbol{H}_{\mathbf{zr}} \overset{\mathsf{SVD}}{=} \boldsymbol{\Gamma}_{\mathbf{z}} \boldsymbol{\Delta}^{(1)} = \begin{bmatrix} \boldsymbol{C}_{\mathbf{z}} \\ \boldsymbol{C}_{\mathbf{z}} \boldsymbol{A}_{11} \\ \vdots \\ \boldsymbol{C}_{\mathbf{z}} \boldsymbol{A}_{11}^{i-1} \end{bmatrix} \begin{bmatrix} \boldsymbol{A}_{11}^{i-1} \boldsymbol{G}^{(1)} & \cdots & \boldsymbol{A}_{11} \boldsymbol{G}^{(1)} & \boldsymbol{G}^{(1)} \end{bmatrix} \tag{9}$$

where $n_1$ is the user-specified dimensionality of the shared latent states $\mathbf{x}_k^{(1)}$, $\boldsymbol{\Gamma}_{\mathbf{z}}$ denotes the observability matrix for the secondary observations, and $\boldsymbol{\Delta}^{(1)}$ denotes the controllability matrix associated with the shared latent states (defined as in equations (3) and (16) in the appendix). At this point, we extract $\boldsymbol{C}_{\mathbf{z}}^{(1)}$ by reading off the first $n_z$ rows of $\boldsymbol{\Gamma}_{\mathbf{z}}$. To extract $\boldsymbol{C}_{\mathbf{r}}^{(1)}$ we first form $\boldsymbol{H}_{\mathbf{r}}$ per equation (2) and extract the observability matrix for $\mathbf{r}$ associated with the shared latent dynamics, $\boldsymbol{\Gamma}_{\mathbf{r}}^{(1)}$, by right multiplying $\boldsymbol{H}_{\mathbf{r}}$ with the pseudoinverse of $\boldsymbol{\Delta}^{(1)}$

$$\boldsymbol{H}_{\mathbf{r}} \boldsymbol{\Delta}^{(1)\dagger} = \boldsymbol{\Gamma}_{\mathbf{r}}^{(1)} = \begin{bmatrix} \boldsymbol{C}_{\mathbf{r}}^{(1)} \\ \boldsymbol{C}_{\mathbf{r}}^{(1)} \boldsymbol{A}_{11} \\ \vdots \\ \boldsymbol{C}_{\mathbf{r}}^{(1)} \boldsymbol{A}_{11}^{i-1} \end{bmatrix}.$$

We then read $\boldsymbol{C}_{\mathbf{r}}^{(1)}$ from the first $n_r$ lines of $\boldsymbol{\Gamma}_{\mathbf{r}}^{(1)}$ (defined as in equation (20) in the appendix). The baseline parameters $\boldsymbol{b}$ and $\boldsymbol{d}$ are empirically computed as the means of $\mathbf{r}_k$ and $\mathbf{z}_k$, respectively.

Lastly, to learn the shared dynamics summarized by the parameter $\boldsymbol{A}_{11}$, we solve the optimization problem $\underline{\boldsymbol{\Delta}}^{(1)} = \boldsymbol{A}_{11}\overline{\boldsymbol{\Delta}}^{(1)}$ where $\underline{\boldsymbol{\Delta}}^{(1)}$ and $\overline{\boldsymbol{\Delta}}^{(1)}$ denote $\boldsymbol{\Delta}^{(1)}$ from which $n_r$ columns have been removed from the right or left, respectively. The closed-form least-squares solution for this problem is $\boldsymbol{A}_{11} = \underline{\boldsymbol{\Delta}}^{(1)}(\overline{\boldsymbol{\Delta}}^{(1)})^{\dagger}$. This concludes the learning of the desired parameters $(\boldsymbol{A}_{11}, \boldsymbol{C}_{\mathbf{r}}^{(1)}, \boldsymbol{C}_{\mathbf{z}}^{(1)}, \boldsymbol{b}, \boldsymbol{d})$, given hyperparameter $n_1$.

### 3.2.2 Stage 2: private dynamics in primary process

After learning the shared dynamics, our algorithm can learn the dynamics private to the primary process that were not captured by $\mathbf{x}_k^{(1)}$. Specifically, we learn the following parameters from equation (7): $([\boldsymbol{A}_{21} \quad \boldsymbol{A}_{22}], \boldsymbol{C}_{\mathbf{r}}^{(2)})$, with hyperparameter $n_2$ determining the unshared latent dimensionality of $\mathbf{r}$. To do so, we first compute a "residual" Hankel matrix, $\boldsymbol{H}_{\mathbf{r}}^{(2)}$, using $\boldsymbol{\Gamma}_{\mathbf{r}}^{(1)}$ and $\boldsymbol{\Delta}^{(1)}$ from stage 1 and decompose it using SVD, keeping the first $n_2$ singular values and vectors

$$\boldsymbol{H}_{\mathbf{r}}^{(2)} = \boldsymbol{H}_{\mathbf{r}} - \boldsymbol{\Gamma}_{\mathbf{r}}^{(1)}\boldsymbol{\Delta}^{(1)} \overset{\mathsf{SVD}}{=} \boldsymbol{\Gamma}_{\mathbf{r}}^{(2)}\boldsymbol{\Delta}^{(2)}. \tag{10}$$

With $\boldsymbol{C}_{\mathbf{r}}^{(2)}$, which corresponds to the first $n_r$ rows of $\boldsymbol{\Gamma}_{\mathbf{r}}^{(2)}$, we construct $\boldsymbol{C}_r = \begin{bmatrix} \boldsymbol{C}_r^{(1)} & \boldsymbol{C}_r^{(2)} \end{bmatrix}$. We then use $\boldsymbol{\Delta}^{(2)}$ to form the controllability matrix $\boldsymbol{\Delta}^{(1,2)}$ as the concatenation of $\boldsymbol{\Delta}^{(1)}$ and $\boldsymbol{\Delta}^{(2)}$ (derivation in appendix A.1):

$$\boldsymbol{\Delta}^{(1,2)} = \begin{bmatrix} \boldsymbol{A}^{(1,2)^{i-1}}\boldsymbol{G}^{(1,2)} & \cdots & \boldsymbol{A}^{(1,2)}\boldsymbol{G}^{(1,2)} & \boldsymbol{G}^{(1,2)} \end{bmatrix} = \begin{bmatrix} \boldsymbol{\Delta}^{(1)} \\ \boldsymbol{\Delta}^{(2)} \end{bmatrix}$$

where $\boldsymbol{A}^{(1,2)}$ refers to the upper left block of the dynamics matrix $\boldsymbol{A}$ that corresponds to the latent states $\mathbf{x}^{(1)}$ and $\mathbf{x}^{(2)}$. Given $\boldsymbol{\Delta}^{(1,2)}$, we extract $\begin{bmatrix} \boldsymbol{A}_{21} & \boldsymbol{A}_{22} \end{bmatrix}$ by solving the problem $\underline{\boldsymbol{\Delta}}^{(2)} = \begin{bmatrix} \boldsymbol{A}_{21} & \boldsymbol{A}_{22} \end{bmatrix}\overline{\boldsymbol{\Delta}}^{(1,2)}$ where

$$\underline{\boldsymbol{\Delta}}^{(2)} := [[\boldsymbol{A}_{21} \quad \boldsymbol{A}_{22}]\,\boldsymbol{A}^{(1,2)^{i-2}}\boldsymbol{G}^{(1,2)} \quad \cdots \quad [\boldsymbol{A}_{21} \quad \boldsymbol{A}_{22}]\,\boldsymbol{G}^{(1,2)}], \quad \overline{\boldsymbol{\Delta}} := [\boldsymbol{A}^{(1,2)^{i-2}}\boldsymbol{G}^{(1,2)} \quad \cdots \quad \boldsymbol{G}^{(1,2)}].$$

Concatenating the sub-blocks together, $\boldsymbol{A}^{(1,2)} = \begin{bmatrix} \boldsymbol{A}_{11} & \boldsymbol{0} \\ \boldsymbol{A}_{21} & \boldsymbol{A}_{22} \end{bmatrix}$. Thus, given hyperparameters $n_1$ and $n_2$, we now have all model parameters associated with the shared dynamics and dynamics private to the primary signal: $(\boldsymbol{A}^{(1,2)}, \boldsymbol{C}_{\mathbf{r}}, \boldsymbol{C}_{\mathbf{z}}^{(1)}, \boldsymbol{b}, \boldsymbol{d})$. The remaining model parameters $(\boldsymbol{A}_{33}, \boldsymbol{C}_{\mathbf{z}}^{(3)})$ are learned in stage 3 (appendices A.1.4 and A.9).

### 3.2.3 Supporting generalized-linear processes

The covariance-based SSID algorithm that we have just derived is what enables our framework to be broadly applicable to generalized-linear time-series data. As discussed in section 2.2, the variables that are linearly related to latent states $\mathbf{x}$ are unobservable in generalized-linear models. However, because the algorithm outlined in sections 3.2.1-3.2.2 only relies on empirical covariances and cross-covariances of the two observation time-series, we can support generalized-linear processes by using moment-conversions (e.g., section 2.2) [21, 24]. When computationally tractable moment conversion equations exist, we can compute both a Hankel matrix $\boldsymbol{H}_{\mathbf{r}}$, as described in section 2.2, and also a cross-term Hankel matrix $\boldsymbol{H}_{\mathbf{zr}}$. For example, for the scenario wherein the Poisson observations constitute the primary process and Gaussian observations the secondary process (i.e., the first four lines of equation (6)), we can compute a moment conversion to estimate joint moments of $\mathbf{z}_k$ and $\mathbf{r}_k$ from the joint moments of the observed signals $\mathbf{z}_k$ and $\mathbf{y}_k$ with the following equation (derived using the conditional statistical properties, see appendix A.1.5)

$$\boldsymbol{\Lambda}_{\mathbf{z}_{f_m}\mathbf{r}_{p_n}} = \mathrm{Cov}(\mathbf{z}_{f_m}, \mathbf{y}_{p_n}) / \boldsymbol{\mu}_{\mathbf{y}_{p_n}} \tag{11}$$

where, similar to equation (5), $m$ and $n$ correspond to indices of the first and second moments of the observation vectors $\mathbf{z}_f$ and $\mathbf{r}_p$ (or $\mathbf{y}_p$), respectively. As another example, if both generalized-linear observations (e.g., $\mathbf{y}$ and $\mathbf{t}$ from equation (6)) are Poisson distributed, the joint moments can be computed as

$$\boldsymbol{\Lambda}_{\mathbf{z}_{f_m}\mathbf{r}_{p_n}} = \ln(\mathrm{Cov}(\mathbf{t}_{f_m}, \mathbf{y}_{p_n}) + \boldsymbol{\mu}_{\mathbf{t}_{f_m}}\boldsymbol{\mu}_{\mathbf{y}_{p_n}}) - \ln(\boldsymbol{\mu}_{\mathbf{t}_{f_m}}\boldsymbol{\mu}_{\mathbf{y}_{p_n}}). \tag{12}$$

For both of these scenarios the baseline log-rates are learned as in PLDSID (section 2.2). Thus, our novel multi-staged covariance-based SSID learning algorithm enables identification of shared vs. private dynamics across various generalized-linear processes.

Table 1: Shared mode identification error (log10, i.e., -2 means 1%) at the shared latent dimensionality ($n_x = n_1$). ✗ indicates that a method (row) does not support the primary observation model (column).

| Method Name | Primary time-series ($\mathbf{r}_k$ or $\mathbf{y}_k$) / Secondary time-series ($\mathbf{z}_k$ or $\mathbf{t}_k$) | | | |
| --- | --- | --- | --- | --- |
| | Gaussian/Gaus. | Poisson/Gaus. | Pois./Pois. | Bernoulli/Gaus. |
| PGLDM (stage 1) | -2.757 ± 0.070 | **-2.707 ± 0.091** | **-1.969 ± 0.079** | **-2.864 ± 0.072** |
| Laplace-EM [26] | -1.320 ± 0.091 | -1.083 ± 0.119 | -1.088 ± 0.110 | -1.027 ± 0.067 |
| PSID (stage 1) [14] | **-2.985 ± 0.102** | ✗ | ✗ | ✗ |
| Covariance SSID [19] | -1.467 ± 0.080 | ✗ | ✗ | ✗ |
| PLDSID [21] | ✗ | -1.319 ± 0.132 | -1.203 ± 0.112 | ✗ |
| bestLDS [24] | ✗ | ✗ | ✗ | -1.209 ± 0.117 |

## 4 Experimental Results

### 4.1 Shared dynamics are accurately identified in generalized-linear simulations

To evaluate how well our method identified the shared dynamics between two generalized-linear time-series, we simulated observations from random dynamical models as per equation (6). All state and observation dimensions were randomly selected, and the corresponding system parameters were randomly generated to simulate stable and slow-decaying dynamics (see appendix A.7.1 for details). In our first experiment, we evaluated how well the shared dynamical subspace could be identified when learning models at the true shared dimensionality. We performed this analysis for four combinations of generalized-linear observation pairs ($\mathbf{r}_k$ or $\mathbf{y}_k$ and $\mathbf{z}_k$ or $\mathbf{t}_k$, respectively): (1) Gaussian/Gaussian, (2) Poisson/Gaussian, (3) Poisson/Poisson, and (4) Bernoulli/Gaussian. Within each configuration, we compared models learned by our method against models learned with either Laplace-EM (expectation-maximization) [26] or a SSID algorithm with the appropriate observation distribution. For the Gaussian/Gaussian case we also compare against PSID [14], an SSID algorithm that preferentially learns the shared dynamics between two Gaussian time-series. PGLDM (our method) and PSID were trained using both the primary and secondary time-series, whereas all other methods used only the primary time-series as they only model a single data source. We evaluated identification of shared dynamics by computing the normalized eigenvalue error between ground truth shared modes (i.e., eigenvalues of $\boldsymbol{A}_{11}$ in equation (7)) and the identified modes (i.e., the learned $\boldsymbol{A}_{11}$ for PGLDM/PSID or $\boldsymbol{A}$ for the other baselines); see appendix A.8.1 for evaluation details. We report the results of this analysis for 20 systems per configuration in Table 1. For almost all conditions PGLDM more accurately identified the shared dynamics.

In our second simulation experiment, we studied the effect of latent state dimension on learning. We generated 16 systems with fixed dimensions for shared and private latent states given by $n_1 = 4, n_2 = 12$, and $n_3 = 4$, accordingly. We swept the learned latent state dimension from 1 to the true dimensionality of the primary observation time-series $n_1 + n_2 = 16$, with the dimensionality of shared dynamics set to $\min(\text{current } n_x, n_1)$. We found that our method accurately identified the shared modes with the minimal latent state dimension of 4; in contrast, PLDSID and Laplace-EM did not reach such high accuracy even when using higher latent state dimensions (figure 1c). In these simulations we also evaluated the predictive power (i.e., correlation coefficient, CC) of the model when using discrete Poisson observations to predict continuous Gaussian observations in a held-out test set (see appendix A.8.2). This second metric allowed us to test our hypothesis that PGLDM's explicit modeling of the shared subspace improved decoding of Gaussian observations from Poisson observations compared with our baselines PLDSID [21] and Laplace-EM [26]. We observed that our method achieved higher decoding performance in low-dimensional regimes, even when using as few as 4 latent states, whereas PLDSID required much larger latent state dimensions (around 12) to reach comparable performance (figure 1a). We also evaluated Poisson self-prediction using area under the receiver operating characteristic curve, AUC (figure 1b). With the inclusion of stage 2 and sufficient model capacity, models learned by PGLDM were able to achieve comparable performance in self-prediction as compared to our baselines.

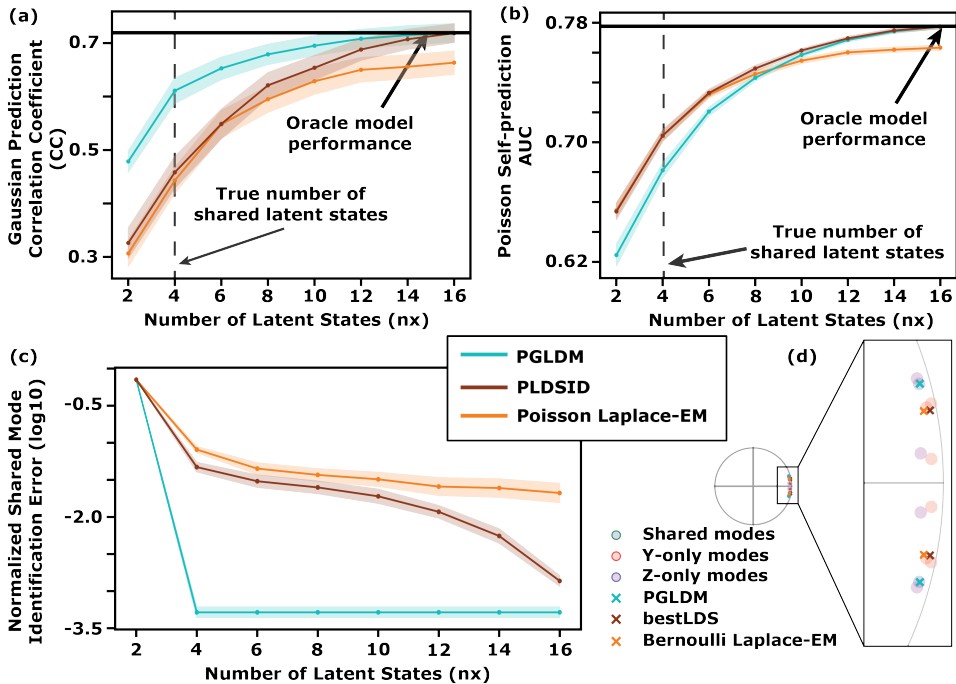

Figure 1: **In simulations, PGLDM more accurately learns the shared dynamical modes and better predicts Gaussian observations from Poisson observations, especially in low-dimensional regimes**. Solid traces show the mean and shaded areas denote the standard error of the mean, (s.e.m.) for each condition. (a-b) Predictive power as a function of latent state dimensionality for all learned models compared against oracle model, i.e., a model with the ground-truth parameters. Left panel (a) shows prediction CC for the Gaussian observations and right panel (b) Poisson self-prediction AUC. (c) The normalized identification error of the shared dynamical modes (in log10 scale) as a function of latent dimensionality. (d) Mode identification with models of size $n_x = n_1 = 2$ for a sample Bernoulli/Gaussian system with true dimensions $n_1 = 2, n_2 = 6, n_3 = 4$.

## 4.2 Modeling shared dynamics improves motor decoding from population spiking activity

As a demonstration on real data, we used PGLDM to model the shared dynamics between discrete Poisson population neural spiking activity and continuous Gaussian arm movements in a publicly available NHP dataset from the Sabes lab [17]. The dataset is of a NHP moving a 2D-cursor in a virtual reality environment based on fingertip position. We use the 2D cursor position and velocity as the continuous observations $\mathbf{z}$. For all methods we used 50ms binned multi-unit spike counts for the discrete observations $\mathbf{y}$. We evaluated decoding performance of learned models using five-fold cross validation across six recording sessions (see appendix A.7.2 for cross-validation details). For PGLDM, we use the shared dynamics dimensionality of $n_1 = \min(\text{current } n_x, 8)$, i.e., a maximum $n_1$ of 8, because behavior decoding using stage 1 roughly plateaued at this dimension.

Compared with PLDSID and Laplace-EM, our method learned models that led to better behavioral decoding at all latent state dimensions, including at the maximum latent state dimension (figure 2a). This result suggests that our method better learns the shared dynamics between Poisson spiking and continuous movement observations due to its ability to dissociate shared vs. private latent states. Interestingly, despite the focus on learning the shared latent states in the first stage, PGLDM was also able to extract the private latent states in the Poisson observations because of its second stage. This stage led to improved neural self-prediction AUC, while still maintaining the more accurate behavioral decoding (figure 2b-c). Indeed, even with the inclusion of just two additional latent states to model private Poisson dynamics ($n_2 = 2, n_x = 10$), neural self-prediction was approaching that of models learned by PLDSID (figure 2b). Finally, given its analytical nature, PGLDM required a substantially lower training time compared with Laplace-EM (see appendix Table 2).

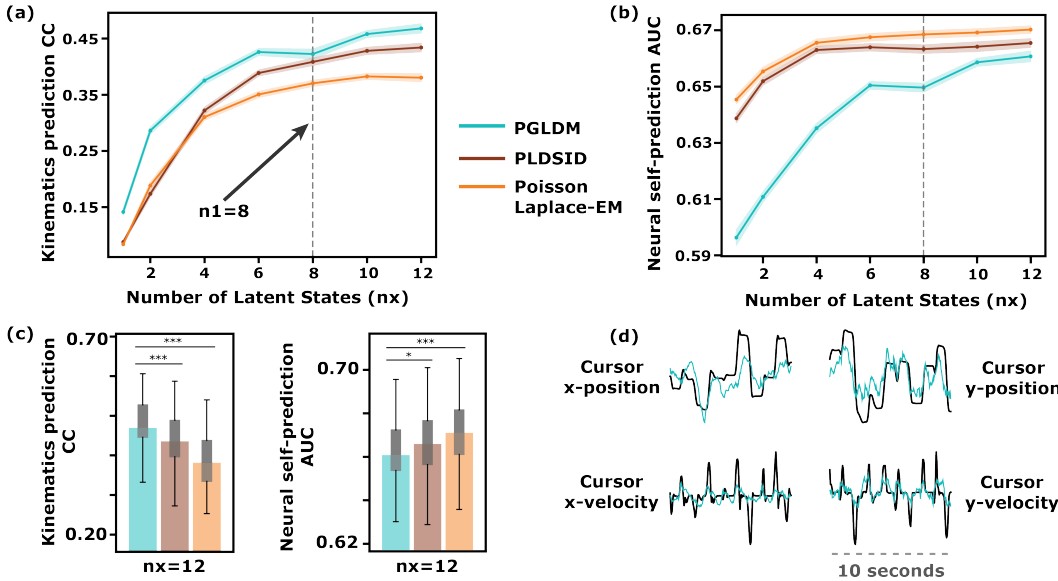

Figure 2: **In NHP data, PGLDM improves movement decoding from Poisson population spiking activity.** (a) Solid traces show the average cross-validated kinematic prediction CC (shaded areas denote the s.e.m.) for models of different latent dimensions learned by PGLDM, PLDSID, and Laplace-EM. (b) Same as (a) but visualizing one-step ahead neural self-prediction AUC. (c) Kinematic prediction CC and neural self-prediction AUC for models of latent dimensionality $n_x = 12$. Asterisks indicate statistical significance (Wilcoxon signed-rank test) with *: $p < 0.05$ and ***: $p < 0.0005$. (d) Example decoding of cursor (x,y) position and velocity from test data.

### 4.3 PGLDM models better decode spiking activity of one visual area from another

As a second demonstration on real data but with a different combination of observation distributions (Poisson/Poisson), we used PGLDM to decode neural population spiking activity in one visual area from another. In a publicly available dataset from Zandvakili and Kohn [16, 18], simultaneous V1/V2 population recordings were performed in anaesthetized NHPs as they were presented visual stimuli. We used five-fold cross validation to evaluate learned model performance in decoding V1 activity from V2 activity and in V2 self-prediction. We again compare with PLDSID and Laplace-EM. For all learning algorithms we tested four latent state dimensions such that $n_x = n_1 + n_2 \in \{2, 4, 6, 8\}$. For PGLDM we used the first two stages, setting the shared dynamics dimensionality to $n_1 = \min(\text{current } n_x, 4)$. Similar to results in figure 2, we chose a maximum $n_1$ of 4 because decoding roughly plateaued at this dimension. The conclusions were consistent with those in figure 2: modeling the shared vs. private dynamics by PGLDM allowed for better decoding of V1 activity while maintaining comparable self-prediction of V2 activity. Analysis details are in appendix A.7.3. We present the results of the complementary analysis (i.e., predicting V1 from V2) in appendix A.12.

### 4.4 Limitations

PGLDM, similar to other SSID methods, uses a time-invariant model which may not be suitable if the data exhibits non-stationarity, such as in chronic neural recordings. To handle non-stationarities, one would need to either intermittently refit the model after a predetermined duration of time, or develop adaptive extensions [27, 28]. As an example of the latter, one can gradually update the model parameters by incorporating a learning rate that weighs recent observations more heavily in the moment computations while gradually forgetting past observations [27]. Moreover, as with other covariance-based SSID methods, PGLDM may be sensitive to the accuracy of the empirical estimates of the first- and second-order moments. However, with increasing number of samples these empirical estimates will approach true statistical values, thereby improving overall performance, as seen in appendix figure 4. Due to errors in the empirical estimates of the covariances, SSID methods may also occasionally learn unstable dynamics (see appendix A.5). Future work may address this by

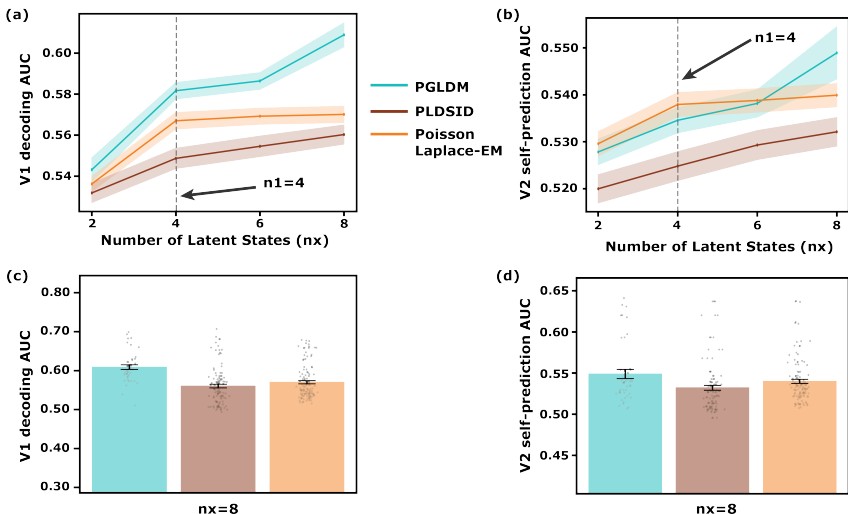

Figure 3: **In NHP data, PGLDM improves V1 decoding from V2 population spiking activity while maintaining comparable V2 self-prediction performance.** (a) Average cross-validated V1 decoding AUC (shaded areas denote the s.e.m.) for models of different latent dimensions. (b) Same as (a) but visualizing V2 one-step ahead self-prediction AUC. (c) V1 decoding AUC at $n_x = 8$. Whiskers correspond to s.e.m. Scatter points are individual trials. (d) Same as (c) but for V2 self-prediction.

incorporating techniques from control theory, such as mode stabilization and covariance matching [29–32]. Finally, although the GLDMs that PGLDM learns are widely used (e.g., in neuroscience), such models may not be suitable for time-series with nonlinearly evolving states. We did not compare our method against nonlinear deep learning methods, such as recurrent neural networks and transformers [33–38], because the goals of these two modeling approaches are different. While nonlinear deep learning methods are typically used to boost overall decoding performance, GLDMs are used for their interpretability and utility in scientific investigations and in real-time, computationally-efficient engineering applications (e.g., brain-computer interfaces).

## 5   Discussion

We developed PGLDM, a novel analytical multi-staged covariance-based SSID algorithm for modeling two generalized-linear processes while also dissociating shared from private dynamics. In simulations we demonstrate that our method successfully achieves this capability agnostic to the generalized-linear observation distribution. As a result, our approach more accurately models system dynamics compared to several commonly-used GLDM variants and their corresponding learning algorithms. We also demonstrate our method's applicability to real data by modeling two distinct NHP datasets recorded under different contexts and from different brain regions. In both simulations and in real data, PGLDM's ability to dissociate shared from private dynamics improved decoding of a secondary time-series from a primary time-series despite using lower-dimensional latent states. Further, although here we specifically focused on modeling Gaussian, Poisson, and Bernoulli observations, our algorithm can be extended to alternate distributions described with generalized-linear models or to other link functions than the ones used here, as long as there exists a corresponding computationally tractable moment conversion equation. This is possible, if a closed-form equation exists, because the covariance-based approach of PGLDM only requires the second-order moments after moment conversion (equations (2), (5), (8), (11), (12)); as such, in these scenarios the moment conversion algorithm can be modified for the desired link function and/or generalized-linear observation model [21, 24]. Beyond neuroscience, due to the high-prevalence of GLDMs across various application domains, our method may be a useful tool for modeling the shared and private dynamics of joint generalized-linear processes with distinct observation distributions.

## Acknowledgments and Disclosure of Funding

This work was supported by Office of Naval Research (ONR) YIP grant N00014-19-1-2128, Army Research Office (ARO) MURI grant W911NF-16-1-0368, and National Institutes of Health (NIH) grants DP2MH126378, R01MH123770, and Brain Initiative R61MH135407. M.M.S. is an inventor on University of Southern California's patents or patent applications related to decoding and closed-loop control approaches, and is a consultant for Paradromics Inc. Authors thank Han-Lin Hsieh, Christian Song, and Parima Ahmadipouranari for helpful discussions.

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

# A  Appendix

## A.1  Derivation

Here we provide the derivation for PGLDM (prioritized generalized-linear dynamical modeling), a covariance-based subspace identification algorithm that learns a dynamical model of a primary time-series while dissociating shared vs. private latents with a secondary time-series. We define the following equivalent formulation for our dynamical model (equation (6)), where the block structure delineates shared ($\mathbf{x}^{(1)}$) and private ($\mathbf{x}^{(2)}, \mathbf{x}^{(3)}$) latent states

$$
\begin{cases}
\begin{bmatrix} \mathbf{x}_{k+1}^{(1)} \\ \mathbf{x}_{k+1}^{(2)} \\ \mathbf{x}_{k+1}^{(3)} \end{bmatrix} = \begin{bmatrix} A_{11} & 0 & 0 \\ A_{21} & A_{22} & 0 \\ 0 & 0 & A_{33} \end{bmatrix} \begin{bmatrix} \mathbf{x}_{k}^{(1)} \\ \mathbf{x}_{k}^{(2)} \\ \mathbf{x}_{k}^{(3)} \end{bmatrix} + \mathbf{w}_k \\[20pt]
\mathbf{r}_k = \begin{bmatrix} C_{\mathbf{r}}^{(1)} & C_{\mathbf{r}}^{(2)} & 0 \end{bmatrix} \begin{bmatrix} \mathbf{x}_{k}^{(1)} \\ \mathbf{x}_{k}^{(2)} \\ \mathbf{x}_{k}^{(3)} \end{bmatrix} + \mathbf{v}_k + \boldsymbol{b} \\[20pt]
\mathbf{z}_k = \begin{bmatrix} C_{\mathbf{z}}^{(1)} & 0 & C_{\mathbf{z}}^{(3)} \end{bmatrix} \begin{bmatrix} \mathbf{x}_{k}^{(1)} \\ \mathbf{x}_{k}^{(2)} \\ \mathbf{x}_{k}^{(3)} \end{bmatrix} + \boldsymbol{\epsilon}_k + \boldsymbol{d} \\[20pt]
\mathbf{y}_k | \mathbf{r}_k \sim \mathcal{P}_{\mathbf{y}|\mathbf{r}}(\mathbf{y}_k; \, g(\mathbf{r}_k)) \\[8pt]
\mathbf{t}_k | \mathbf{z}_k \sim \mathcal{P}_{\mathbf{t}|\mathbf{z}}(\mathbf{t}_k; \, h(\mathbf{z}_k))
\end{cases}
\tag{13}
$$

with parameters and noise terms defined as in sections 2.1-2.2 and 3.1.

### A.1.1  Standard Covariance-Based SSID

Before we present the derivation for PGLDM, we review a few steps in standard covariance-based SSID (section 2.1) that will help us in the derivation. First, it can be shown that the $\tau$-th lag cross-covariance terms for $\mathbf{r}$ can be written in terms of model parameters as $\Lambda_{\mathbf{r}_\tau} = \mathrm{Cov}\,(\mathbf{r}_{k+\tau}, \mathbf{r}_k) = C_{\mathbf{r}} A^{\tau-1} G$, where $G := \mathrm{Cov}(\mathbf{x}_{k+1}, \mathbf{r}_k)$. Using this relationship, the Hankel matrix, $H_{\mathbf{r}}$, can be expanded as [19, 20]

$$
\begin{aligned}
H_{\mathbf{r}} = \mathrm{Cov}(\mathbf{r}_f, \mathbf{r}_p) &= \begin{bmatrix} \Lambda_{\mathbf{r}_i} & \Lambda_{\mathbf{r}_{i-1}} & \cdots & \Lambda_{\mathbf{r}_1} \\ \vdots & \vdots & \cdots & \vdots \\ \Lambda_{\mathbf{r}_{2i-1}} & \Lambda_{\mathbf{r}_{2i-2}} & \cdots & \Lambda_{\mathbf{r}_i} \end{bmatrix} \\[10pt]
&= \begin{bmatrix} C_{\mathbf{r}} A^{i-1} G & C_{\mathbf{r}} A^{i-2} G & \cdots & C_{\mathbf{r}} G \\ \vdots & \vdots & \cdots & \vdots \\ C_{\mathbf{r}} A^{2i-2} G & C_{\mathbf{r}} A^{2i-3} G & \cdots & C_{\mathbf{r}} A^{i-1} G \end{bmatrix}.
\end{aligned}
\tag{14}
$$

Second, using a singular-value decomposition the above Hankel matrix $H_{\mathbf{r}}$ can be decomposed into observability, $\Gamma_{\mathbf{r}}$, and controllability, $\Delta$, matrices from which model parameters can be extracted [19, 20]

$$
H_{\mathbf{r}} \overset{\text{SVD}}{=} \Gamma_{\mathbf{r}} \Delta = \begin{bmatrix} C_{\mathbf{r}} \\ C_{\mathbf{r}} A \\ \vdots \\ C_{\mathbf{r}} A^{i-1} \end{bmatrix} \begin{bmatrix} A^{i-1} G & \cdots & AG & G \end{bmatrix}.
\tag{15}
$$

We note that there exists a canonical correlation analysis (CCA) version of this algorithm wherein $H_{\mathbf{r}}$ is left and right normalized by the square root matrices of the future and past observation covariance

matrices, respectively, prior to the singular-value decomposition. Specifically $L^{-1} \boldsymbol{H_r} M^{-T} \stackrel{\text{SVD}}{=} \hat{U} \hat{\Sigma} \hat{V}^T$, where $\boldsymbol{\Lambda}_{ff} = \text{Cov}(\mathbf{r}_f, \mathbf{r}_f) = LL^T$ and $\boldsymbol{\Lambda}_{pp} = \text{Cov}(\mathbf{r}_p, \mathbf{r}_p) = MM^T$. The observability and controllability matrices can be recovered as $\boldsymbol{\Gamma_r} = L\hat{U}\hat{\Sigma}^{1/2}$ and $\boldsymbol{\Delta} = \hat{\Sigma}^{1/2}\hat{V}^T M^T$, respectively.

We refer readers to section 8.7 of Katayama [20] for more detail. We can similarly derive a CCA version of PGLDM, which we present next. This can be achieved by applying the appropriate normalizations to both $\boldsymbol{H_r}$ and $\boldsymbol{H_{zr}}$, the cross-term Hankel matrix defined below, prior to the singular-value decomposition. The resulting decomposition would then have to be un-normalized to retrieve the appropriate observability and controllability matrices.

### A.1.2 PGLDM: Stage 1 derivation

In the first stage of our algorithm, our goal is to learn the model parameters that correspond to the shared dynamical subspace of $\mathbf{z}$ and $\mathbf{r}$ via the latent state $\mathbf{x}_k^{(1)}$: $(\boldsymbol{A}_{11}, \boldsymbol{C_r}^{(1)}, \boldsymbol{C_z}^{(1)}, \boldsymbol{b}, \boldsymbol{d})$. Before deriving the first stage, we first present a few parameter definitions and simplifying assumptions. As noted in section 3.1, $\boldsymbol{G}$ can be partitioned as $\boldsymbol{G} = \begin{bmatrix} \boldsymbol{G}^{(1)^T} & \boldsymbol{G}^{(2)^T} & \boldsymbol{G}^{(3)^T} \end{bmatrix}^T$ due to the block structure of equation (13). Further, based on assumptions 3.2 and 3.3 we can set $\boldsymbol{G}^{(3)} = 0$. Leveraging the block structure of $\boldsymbol{G}$ we can also define a simplified block-partitioned structure for the controllability matrix $\boldsymbol{\Delta}$ (equation (15)) as

$$
\begin{aligned}
\boldsymbol{\Delta} &= \begin{bmatrix} \boldsymbol{A}^{i-1}\boldsymbol{G} & \cdots & \boldsymbol{A}\boldsymbol{G} & \boldsymbol{G} \end{bmatrix} \\[2mm]
&= \begin{bmatrix} \begin{bmatrix} \boldsymbol{A}_{11} & \boldsymbol{0} & \boldsymbol{0} \\ \boldsymbol{A}_{21} & \boldsymbol{A}_{22} & \boldsymbol{0} \\ \boldsymbol{0} & \boldsymbol{0} & \boldsymbol{A}_{33} \end{bmatrix}\boldsymbol{A}^{i-2}\boldsymbol{G} & \cdots & \begin{bmatrix} \boldsymbol{A}_{11} & \boldsymbol{0} & \boldsymbol{0} \\ \boldsymbol{A}_{21} & \boldsymbol{A}_{22} & \boldsymbol{0} \\ \boldsymbol{0} & \boldsymbol{0} & \boldsymbol{A}_{33} \end{bmatrix}\begin{bmatrix} \boldsymbol{G}^{(1)} \\ \boldsymbol{G}^{(2)} \\ \boldsymbol{G}^{(3)} \end{bmatrix} & \begin{bmatrix} \boldsymbol{G}^{(1)} \\ \boldsymbol{G}^{(2)} \\ \boldsymbol{G}^{(3)} \end{bmatrix} \end{bmatrix} \\[2mm]
&= \begin{bmatrix} \begin{bmatrix} \boldsymbol{A}_{11}^{i-1}\boldsymbol{G}^{(1)} \\ [\boldsymbol{A}_{21} \ \ \boldsymbol{A}_{22} \ \ \boldsymbol{0}]\boldsymbol{A}^{i-2}\boldsymbol{G} \\ \boldsymbol{A}_{33}^{i-1}\boldsymbol{G}^{(3)} \end{bmatrix} & \cdots & \begin{bmatrix} \boldsymbol{A}_{11}\boldsymbol{G}^{(1)} \\ [\boldsymbol{A}_{21} \ \ \boldsymbol{A}_{22} \ \ \boldsymbol{0}]\boldsymbol{G} \\ \boldsymbol{A}_{33}\boldsymbol{G}^{(3)} \end{bmatrix} & \begin{bmatrix} \boldsymbol{G}^{(1)} \\ \boldsymbol{G}^{(2)} \\ \boldsymbol{G}^{(3)} \end{bmatrix} \end{bmatrix} = \begin{bmatrix} \boldsymbol{\Delta}^{(1)} \\ \boldsymbol{\Delta}^{(2)} \\ \boldsymbol{\Delta}^{(3)} \end{bmatrix} \quad (16) \\[2mm]
&= \begin{bmatrix} \begin{bmatrix} \boldsymbol{A}_{11}^{i-1}\boldsymbol{G}^{(1)} \\ [\boldsymbol{A}_{21} \ \ \boldsymbol{A}_{22} \ \ \boldsymbol{0}]\boldsymbol{A}^{i-2}\boldsymbol{G} \\ \boldsymbol{0} \end{bmatrix} & \cdots & \begin{bmatrix} \boldsymbol{A}_{11}\boldsymbol{G}^{(1)} \\ [\boldsymbol{A}_{21} \ \ \boldsymbol{A}_{22}]\begin{bmatrix} \boldsymbol{G}^{(1)} \\ \boldsymbol{G}^{(2)} \end{bmatrix} \\ \boldsymbol{0} \end{bmatrix} & \begin{bmatrix} \boldsymbol{G}^{(1)} \\ \boldsymbol{G}^{(2)} \\ \boldsymbol{0} \end{bmatrix} \end{bmatrix} = \begin{bmatrix} \boldsymbol{\Delta}^{(1)} \\ \boldsymbol{\Delta}^{(2)} \\ \boldsymbol{0} \end{bmatrix}.
\end{aligned}
$$

The block-partition structure of $\boldsymbol{\Delta}$ will play an important role in the separation of learning between stages 1 and 2 and will enable the prioritized identification of shared dynamics $\boldsymbol{A}_{11}$.

To begin the derivation, it can be shown that the $\tau$-th lag cross-covariance between $\mathbf{z}$ and $\mathbf{r}$ can be written in terms of model parameters as $\boldsymbol{\Lambda}_{\mathbf{zr}_\tau} = \text{Cov}(\mathbf{z}_{k+\tau}, \mathbf{r}_k) = \boldsymbol{C_z}\boldsymbol{A}^{\tau-1}\boldsymbol{G}$. Using the block definition of $\boldsymbol{G}$, we can simplify the cross-covariance term as

$$
\boldsymbol{\Lambda}_{\mathbf{zr}_\tau} = \begin{bmatrix} \boldsymbol{C_z}^{(1)} & \boldsymbol{0} & \boldsymbol{C_z}^{(3)} \end{bmatrix} \begin{bmatrix} \boldsymbol{A}_{11} & \boldsymbol{0} & \boldsymbol{0} \\ \boldsymbol{A}_{21} & \boldsymbol{A}_{22} & \boldsymbol{0} \\ \boldsymbol{0} & \boldsymbol{0} & \boldsymbol{A}_{33} \end{bmatrix}^{\tau-1} \begin{bmatrix} \boldsymbol{G}^{(1)} \\ \boldsymbol{G}^{(2)} \\ \boldsymbol{G}^{(3)} \end{bmatrix} = \boldsymbol{C_z}^{(1)}\boldsymbol{A}_{11}^{\tau-1}\boldsymbol{G}^{(1)}.
$$

The Hankel matrix between future secondary observations and past primary observations can then be expanded as

$$
\begin{aligned}
\boldsymbol{H_{zr}} = \text{Cov}(\mathbf{z}_f, \mathbf{r}_p) &= \begin{bmatrix} \boldsymbol{\Lambda}_{\mathbf{zr}_i} & \boldsymbol{\Lambda}_{\mathbf{zr}_{i-1}} & \cdots & \boldsymbol{\Lambda}_{\mathbf{zr}_1} \\ \vdots & \vdots & \cdots & \vdots \\ \boldsymbol{\Lambda}_{\mathbf{zr}_{2i-1}} & \boldsymbol{\Lambda}_{\mathbf{zr}_{2i-2}} & \cdots & \boldsymbol{\Lambda}_{\mathbf{zr}_i} \end{bmatrix} \\[2mm]
&= \begin{bmatrix} \boldsymbol{C_z}^{(1)}\boldsymbol{A}_{11}^{i-1}\boldsymbol{G}^{(1)} & \boldsymbol{C_z}^{(1)}\boldsymbol{A}_{11}^{i-2}\boldsymbol{G}^{(1)} & \cdots & \boldsymbol{C_z}^{(1)}\boldsymbol{G}^{(1)} \\ \vdots & \vdots & \cdots & \vdots \\ \boldsymbol{C_z}^{(1)}\boldsymbol{A}_{11}^{2i-2}\boldsymbol{G}^{(1)} & \boldsymbol{C_z}^{(1)}\boldsymbol{A}_{11}^{2i-3}\boldsymbol{G}^{(1)} & \cdots & \boldsymbol{C_z}^{(1)}\boldsymbol{A}_{11}^{i-1}\boldsymbol{G}^{(1)} \end{bmatrix}.
\end{aligned}
$$

A singular-value decomposition of $\boldsymbol{H_{zr}}$ yields the observability matrix for $\mathbf{z}$ (i.e., $\boldsymbol{\Gamma_z}$) and the controllability matrix $\boldsymbol{\Delta}^{(1)}$ associated with the shared dynamics

$$\boldsymbol{H_{zr}} \overset{\text{SVD}}{=} \boldsymbol{\Gamma_z}\boldsymbol{\Delta}^{(1)} = \begin{bmatrix} \boldsymbol{C_z}^{(1)} \\ \boldsymbol{C_z}^{(1)}\boldsymbol{A}_{11} \\ \vdots \\ \boldsymbol{C_z}^{(1)}\boldsymbol{A}_{11}^{i-1} \end{bmatrix} \begin{bmatrix} \boldsymbol{A}_{11}^{i-1}\boldsymbol{G}^{(1)} & \cdots & \boldsymbol{A}_{11}\boldsymbol{G}^{(1)} & \boldsymbol{G}^{(1)} \end{bmatrix}. \tag{17}$$

At this point, $\boldsymbol{C_z}^{(1)}$ can be read off the first $n_z$ rows of $\boldsymbol{\Gamma_z}$. The shared latent dynamics matrix $\boldsymbol{A}_{11}$ can be learned by solving a least-squares problem based on the controllability matrix $\boldsymbol{\Delta}^{(1)}$ (as introduced in section 3.2.1)

$$\underline{\boldsymbol{\Delta}}^{(1)} = \boldsymbol{A}_{11}\overline{\boldsymbol{\Delta}}^{(1)} \quad \text{where} \tag{18}$$

$$\underline{\boldsymbol{\Delta}}^{(1)} := \begin{bmatrix} \boldsymbol{A}_{11}^{i-1}\boldsymbol{G}^{(1)} & \cdots & \boldsymbol{A}_{11}\boldsymbol{G}^{(1)} \end{bmatrix}, \quad \overline{\boldsymbol{\Delta}}^{(1)} := \begin{bmatrix} \boldsymbol{A}_{11}^{i-2}\boldsymbol{G}^{(1)} & \cdots & \boldsymbol{G}^{(1)} \end{bmatrix},$$

which has the following closed-form solution: $\boldsymbol{A}_{11} = \underline{\boldsymbol{\Delta}}^{(1)}(\overline{\boldsymbol{\Delta}}^{(1)})^{\dagger}$.

To extract $\boldsymbol{C_r}^{(1)}$, we first note that the Hankel expansion in equation (15) can be rewritten with the block-structure of $\boldsymbol{\Delta}$ in mind as (we omit partitions corresponding to latents $x^{(3)}$ due to the decoupling)

$$\begin{aligned}
\boldsymbol{H_r} = \mathbf{U}\boldsymbol{\Sigma}\mathbf{V}^T \quad &= \overset{\boldsymbol{\Gamma_r}}{(\mathbf{U}\boldsymbol{\Sigma}^{1/2})}\overset{\boldsymbol{\Delta}}{(\boldsymbol{\Sigma}^{1/2}\mathbf{V}^T)} = \overset{\boldsymbol{\Gamma_r}}{(\mathbf{U}\boldsymbol{\Sigma}^{1/2})}\overset{\begin{bmatrix}\boldsymbol{\Delta}^{(1)}\\\boldsymbol{\Delta}^{(2)}\end{bmatrix}}{(\boldsymbol{\Sigma}^{1/2}\mathbf{V}^T)} \\[2mm]
&\overset{(a)}{=} \left( \begin{bmatrix} \mathbf{U}^{(1)} & \mathbf{U}^{(2)} \end{bmatrix} \begin{bmatrix} \boldsymbol{\Sigma}^{(1)^{1/2}} & \mathbf{0} \\ \mathbf{0} & \boldsymbol{\Sigma}^{(2)^{1/2}} \end{bmatrix} \right) \left( \begin{bmatrix} \boldsymbol{\Sigma}^{(1)^{1/2}} & \mathbf{0} \\ \mathbf{0} & \boldsymbol{\Sigma}^{(2)^{1/2}} \end{bmatrix} \begin{bmatrix} \mathbf{V}^{(1)^T} \\ \mathbf{V}^{(2)^T} \end{bmatrix} \right) \\[2mm]
&= (\mathbf{U}^{(1)}\boldsymbol{\Sigma}^{(1)^{1/2}})\overset{\boldsymbol{\Delta}^{(1)}}{(\boldsymbol{\Sigma}^{(1)^{1/2}}\mathbf{V}^{(1)^T})} + (\mathbf{U}^{(2)}\boldsymbol{\Sigma}^{(2)^{1/2}})\overset{\boldsymbol{\Delta}^{(2)}}{(\boldsymbol{\Sigma}^{(2)^{1/2}}\mathbf{V}^{(2)^T})} \\[2mm]
&\overset{(b)}{=} \boldsymbol{\Gamma_r}^{(1)}\boldsymbol{\Delta}^{(1)} + \boldsymbol{\Gamma_r}^{(2)}\boldsymbol{\Delta}^{(2)}
\end{aligned} \tag{19}$$

where equivalence (a) is due to the block-partition structure of $\boldsymbol{\Delta}$ and equivalence (b) implicitly introduces a block structure to observability matrix $\boldsymbol{\Gamma_r}$, where $\boldsymbol{\Gamma_r}^{(1)}$ and $\boldsymbol{\Gamma_r}^{(2)}$ correspond to the observability matrices associated with the shared and private latents, respectively. More formally,

$$\boldsymbol{\Gamma_r} = \begin{bmatrix} \boldsymbol{\Gamma_r}^{(1)} & \boldsymbol{\Gamma_r}^{(2)} \end{bmatrix} = \begin{bmatrix} \boldsymbol{C_r} \\ \boldsymbol{C_r}\boldsymbol{A} \\ \vdots \\ \boldsymbol{C_r}\boldsymbol{A}^{i-1} \end{bmatrix} = \begin{bmatrix} \boldsymbol{C_r}^{(1)} & \boldsymbol{C_r}^{(2)} \\ \overline{\boldsymbol{\Gamma}}_{\mathbf{r}}^{(1)} & \overline{\boldsymbol{\Gamma}}_{\mathbf{r}}^{(2)} \end{bmatrix}, \tag{20}$$

where $\overline{\boldsymbol{\Gamma}}_{\mathbf{r}}$ denotes $\boldsymbol{\Gamma_r}$ from which the top $n_r$ rows have been removed. Taken together, $\boldsymbol{H_r}$ can be viewed as the sum of "shared" and "private" components (equation (19)). Thus, we can compute $\boldsymbol{\Gamma_r}^{(1)}$ as

$$\boldsymbol{H_r}\boldsymbol{\Delta}^{(1)\dagger} = (\boldsymbol{\Gamma_r}^{(1)}\boldsymbol{\Delta}^{(1)} + \boldsymbol{\Gamma_r}^{(2)}\boldsymbol{\Delta}^{(2)})\boldsymbol{\Delta}^{(1)\dagger} = \boldsymbol{\Gamma_r}^{(1)} \tag{21}$$

where we have used the orthonormal property of right singular vectors $\mathbf{V}$ to conclude $\boldsymbol{\Delta}^{(2)}\boldsymbol{\Delta}^{(1)\dagger} = \mathbf{0}$. At this point, we can extract $\boldsymbol{C_r}^{(1)}$ by reading the top $n_r$ rows of $\boldsymbol{\Gamma_r}^{(1)}$. Finally, $\boldsymbol{b}$ and $\boldsymbol{d}$ can both be learned directly from the data either by computing the empirical mean (when working with continuous Gaussian observations) or during the moment transformation (section 2.2). This concludes the learning of all parameters associated with the shared dynamical subspace, i.e., $(\boldsymbol{A}_{11}, \boldsymbol{C_r}^{(1)}, \boldsymbol{C_z}^{(1)}, \boldsymbol{b}, \boldsymbol{d})$.

### A.1.3  PGLDM: Stage 2 derivation

In the second stage of our algorithm, our goal is to learn model parameters that describe the private dynamics of $\mathbf{r}$ via the latent state $\mathbf{x}_k^{(2)}$: $([\boldsymbol{A}_{21} \quad \boldsymbol{A}_{22}], \boldsymbol{C_r}^{(2)})$. To learn these parameters, we first extract the private component in equation (19), termed $\boldsymbol{H_r}^{(2)}$, by subtracting $\boldsymbol{\Gamma_r}^{(1)}\boldsymbol{\Delta}^{(1)}$ from $\boldsymbol{H_r}$,

and decompose it via a singular-value decomposition to get $\mathbf{\Gamma}_{\mathbf{r}}^{(2)}$ and $\mathbf{\Delta}^{(2)}$ as (we omit reference to private latents $\mathbf{x}^{(3)}$ due to the decoupling)

$$H_{\mathbf{r}}^{(2)} = H_{\mathbf{r}} - \mathbf{\Gamma}_{\mathbf{r}}^{(1)}\mathbf{\Delta}^{(1)} \overset{\mathsf{SVD}}{=} \mathbf{\Gamma}_{\mathbf{r}}^{(2)}\mathbf{\Delta}^{(2)}. \tag{22}$$

At this point, we take $C_{\mathbf{r}}^{(2)}$ as the top $n_r$ rows of $\mathbf{\Gamma}_{\mathbf{r}}^{(2)}$ and concatentate with $C_{\mathbf{r}}^{(1)}$ to complete $C_{\mathbf{r}}$.

To complete the state dynamics matrix $A$, we refer back to the block-structure representation of the controllability matrix in equation (16)

$$\begin{bmatrix} \mathbf{\Delta}^{(1)} \\ \mathbf{\Delta}^{(2)} \end{bmatrix} = \left[ \begin{bmatrix} A_{11} & 0 \\ A_{21} & A_{22} \end{bmatrix} A^{i-2}G \quad \cdots \quad \begin{bmatrix} A_{11} & 0 \\ A_{21} & A_{22} \end{bmatrix}\begin{bmatrix} G^{(1)} \\ G^{(2)} \end{bmatrix} \quad \begin{bmatrix} G^{(1)} \\ G^{(2)} \end{bmatrix} \right]$$

from which we construct the following relationship

$$\begin{bmatrix} \mathbf{\Delta}^{(1)} \\ \mathbf{\Delta}^{(2)} \end{bmatrix} = \begin{bmatrix} A_{11} & 0 \\ A_{21} & A_{22} \end{bmatrix}\begin{bmatrix} \overline{\mathbf{\Delta}}^{(1)} \\ \overline{\mathbf{\Delta}}^{(2)} \end{bmatrix} \tag{23}$$

where $\overline{\mathbf{\Delta}}$ and $\underline{\mathbf{\Delta}}$ are defined as in equation (18). We can further isolate the residual state transitions as the solution to the following equation (taken from the second row of equation (23))

$$\underline{\mathbf{\Delta}}^{(2)} = [A_{21} \quad A_{22}]\begin{bmatrix} \overline{\mathbf{\Delta}}^{(1)} \\ \overline{\mathbf{\Delta}}^{(2)} \end{bmatrix} = [A_{21} \quad A_{22}]\overline{\mathbf{\Delta}}, \tag{24}$$

which has the following closed-form least-squares solution: $[A_{21} \quad A_{22}] = \underline{\mathbf{\Delta}}^{(2)}\overline{\mathbf{\Delta}}^{\dagger}$. The full state dynamics is the concatenation $A = \begin{bmatrix} A_{11} & 0 \\ A_{21} & A_{22} \end{bmatrix}$. This concludes the learning of parameters for the private dynamics in $\mathbf{r}$, i.e., $([A_{21} \quad A_{22}], C_{\mathbf{r}}^{(2)})$.

### A.1.4 PGLDM: Stage 3 derivation

Finally, in the third state we can learn the model parameters associated with the private dynamics in $\mathbf{z}$, that is $(A_{33}, C_{\mathbf{z}}^{(3)})$, in an approach similar to stage 2. We first construct a future-past Hankel matrix, $H_{\mathbf{z}}$, associated with the secondary observation $\mathbf{z}$. Using an analysis similar to equations (19) and (21), we can show that $H_{\mathbf{z}}$ as the sum of "shared" and "private" components. In the case of secondary observation $\mathbf{z}$, we can explicitly show that its observability matrix, $\mathbf{\Gamma}_{\mathbf{z}}$, assumes a block form as

$$\mathbf{\Gamma}_{\mathbf{z}} = \begin{bmatrix} C_{\mathbf{z}} \\ C_{\mathbf{z}}A \\ \vdots \\ C_{\mathbf{z}}A^{i-1} \end{bmatrix} \overset{(a)}{=} \begin{bmatrix} C_{\mathbf{z}}^{(1)} & 0 & C_{\mathbf{z}}^{(3)} \\ C_{\mathbf{z}}^{(1)}A_{11} & 0 & C_{\mathbf{z}}^{(3)}A_{33} \\ \vdots & & \vdots \\ C_{\mathbf{z}}^{(1)}A_{11}^{i-1} & 0 & C_{\mathbf{z}}^{(3)}A_{33}^{i-1} \end{bmatrix} = \begin{bmatrix} \mathbf{\Gamma}_{\mathbf{z}}^{(1)} & 0 & \mathbf{\Gamma}_{\mathbf{z}}^{(3)} \end{bmatrix}, \tag{25}$$

where equivalence (a) is due to the block-partition definitions of $C_{\mathbf{z}}$ and $A$ in equation (13). Henceforth, we omit the middle block of $\mathbf{\Gamma}_{\mathbf{z}}$, without any loss of generality, and simplify $H_{\mathbf{z}}$ as

$$\begin{aligned} H_{\mathbf{z}} = \mathbf{U\Sigma V}^T \quad &= (\overset{\mathbf{\Gamma}_{\mathbf{z}}}{\mathbf{U\Sigma}^{1/2}})(\overset{\mathbf{\Delta}_{\mathbf{z}}}{\mathbf{\Sigma}^{1/2}\mathbf{V}^T}) = (\overset{[\mathbf{\Gamma}_{\mathbf{z}}^{(1)} \quad \mathbf{\Gamma}_{\mathbf{z}}^{(3)}]}{\mathbf{U\Sigma}^{1/2}}) (\overset{\mathbf{\Delta}_{\mathbf{z}}}{\mathbf{\Sigma}^{1/2}\mathbf{V}^T}) \\[2mm] &\overset{(a)}{=} \left( \begin{bmatrix} \mathbf{U}^{(1)} & \mathbf{U}^{(3)} \end{bmatrix} \begin{bmatrix} \mathbf{\Sigma}^{(1)^{1/2}} & 0 \\ 0 & \mathbf{\Sigma}^{(3)^{1/2}} \end{bmatrix} \right) \left( \begin{bmatrix} \mathbf{\Sigma}^{(1)^{1/2}} & 0 \\ 0 & \mathbf{\Sigma}^{(3)^{1/2}} \end{bmatrix} \begin{bmatrix} \mathbf{V}^{(1)^T} \\ \mathbf{V}^{(3)^T} \end{bmatrix} \right) \\[2mm] &= (\overset{\mathbf{\Gamma}_{\mathbf{z}}^{(1)}}{\mathbf{U}^{(1)}\mathbf{\Sigma}^{(1)^{1/2}}})(\mathbf{\Sigma}^{(1)^{1/2}}\mathbf{V}^{(1)^T}) + (\overset{\mathbf{\Gamma}_{\mathbf{z}}^{(3)}}{\mathbf{U}^{(3)}\mathbf{\Sigma}^{(3)^{1/2}}})(\mathbf{\Sigma}^{(3)^{1/2}}\mathbf{V}^{(3)^T}) \\[2mm] &\overset{(b)}{=} \mathbf{\Gamma}_{\mathbf{z}}^{(1)}\mathbf{\Delta}_{\mathbf{z}}^{(1)} + \mathbf{\Gamma}_{\mathbf{z}}^{(3)}\mathbf{\Delta}_{\mathbf{z}}^{(3)} \end{aligned} \tag{26}$$

where equivalence (a) used the block structure of $\mathbf{\Gamma}_{\mathbf{z}}$ and equivalence (b) implicitly introduced a block structure on the controllability matrix associated with observation $\mathbf{z}$. In stage 1 we had learned

$\mathbf{\Gamma}_{\mathbf{z}}^{(1)}$, thus we can use an approach similar to equations (21)-(22) to extract the private component of $\mathbf{H}_{\mathbf{z}}$. We first compute $\mathbf{\Delta}_{\mathbf{z}}^{(1)}$, the $\mathbf{z}$ controllabilty matrix associated with the shared dynamics, as

$$\mathbf{\Gamma}_{\mathbf{z}}^{(1)\dagger} \mathbf{H}_{\mathbf{z}} = \mathbf{\Gamma}_{\mathbf{z}}^{(1)\dagger}(\mathbf{\Gamma}_{\mathbf{z}}^{(1)}\mathbf{\Delta}_{\mathbf{z}}^{(1)} + \mathbf{\Gamma}_{\mathbf{z}}^{(3)}\mathbf{\Delta}_{\mathbf{z}}^{(3)}) = \mathbf{\Delta}_{\mathbf{z}}^{(1)}. \tag{27}$$

Then we can extract and decompose the private component of the Hankel matrix

$$\mathbf{H}_{\mathbf{z}}^{(3)} = \mathbf{H}_{\mathbf{z}} - \mathbf{\Gamma}_{\mathbf{z}}^{(1)}\mathbf{\Delta}_{\mathbf{z}}^{(1)} \stackrel{\mathsf{SVD}}{=} \mathbf{\Gamma}_{\mathbf{z}}^{(3)}\mathbf{\Delta}_{\mathbf{z}}^{(3)}. \tag{28}$$

As in the previous stages, $\mathbf{C}_{\mathbf{z}}^{(3)}$ is taken as the top $n_z$ rows of $\mathbf{\Gamma}_{\mathbf{z}}^{(3)}$, thus completing $\mathbf{C}_{\mathbf{z}}$. To learn $\mathbf{A}_{33}$ we use the same approach as standard covariance-based SSID (section 2.1) and solve the problem $\overline{\mathbf{\Gamma}}_{\mathbf{z}}^{(3)} = \underline{\mathbf{\Gamma}}_{\mathbf{z}}^{(3)}\mathbf{A}_{33}$, where $\overline{\mathbf{\Gamma}}_{\mathbf{z}}$ and $\underline{\mathbf{\Gamma}}_{\mathbf{z}}$ denote $\mathbf{\Gamma}_{\mathbf{z}}$ from which the top or bottom $n_z$ rows have been removed, respectively. This optimization problem has the closed-form least-squares solution $\mathbf{A}_{33} = \underline{\mathbf{\Gamma}}_{\mathbf{z}}^{\dagger}\overline{\mathbf{\Gamma}}_{\mathbf{z}}$. This concludes the learning of parameters for the private dynamics in $\mathbf{z}$, i.e., $(\mathbf{A}_{33}, \mathbf{C}_{\mathbf{z}}^{(3)})$.

### A.1.5 Transformation of joint Gaussian and Poisson moments

In sections A.1.1-A.1.4 we demonstrated how all model parameters can be extracted in three stages with prioritization, starting from the second-moments of the $\mathbf{r}$ and $\mathbf{z}$. As an example of how these moments can be computed from generalized-linear observations, we consider the specific case of Poisson/Gaussian observations as $\mathbf{y}$ and $\mathbf{z}$, respectively. The joint second-moment $\mathbf{\Lambda}_{\mathbf{z}_{f_m}\mathbf{r}_{p_n}}$ can be analytically recovered from the computable moments of $\mathbf{y}$ (the Poisson observations) and $\mathbf{z}$ (the Gaussian observations), using equation (11) as

$$\mathbf{\Lambda}_{\mathbf{z}_{f_m}\mathbf{r}_{p_n}} = \mathrm{Cov}(\mathbf{z}_{f_m}, \mathbf{y}_{p_n}) / \boldsymbol{\mu}_{\mathbf{y}_{p_n}}$$

where $m$ and $n$ indicate index-wise notation. Here we provide a sketch of the proof. Without loss of generality, assume $\mathbf{z}$ and $\mathbf{r}$ are stationary with a mean of $\mathbf{0}$ (e.g., demeaned during preprocessing). We can compute the covariance of any two elements $j$ and $k$ of vectors $\mathbf{z}_f$ and $\mathbf{y}_p$ respectively as

$$\mathrm{Cov}(\mathbf{z}_{f_j}, \mathbf{y}_{p_k}) \quad = E\left[\mathbf{z}_{f_j}\mathbf{y}_{p_k}\right] = E\left[E\left[\mathbf{z}_{f_j}\mathbf{y}_{p_k}|\mathbf{r}_{p_k}\right]\right]$$

$$\stackrel{(a)}{=} E\left[E\left[\mathbf{z}_{f_j}|\mathbf{r}_{p_k}\right] E\left[\mathbf{y}_{f_k}|\mathbf{r}_{p_k}\right]\right] = E\left[E\left[\mathbf{z}_{f_j}|\mathbf{r}_{p_k}\right] \exp(\mathbf{r}_{p_k})\right]$$

where (a) is because $\mathbf{z}_{f_j}$ and $\mathbf{y}_{p_k}$ are independent when conditioned on latent log-rate $\mathbf{r}_{p_k}$. Next, we use the fact that $\mathbf{z}_f$ and $\mathbf{r}_p$ are jointly Gaussian random processes and, as a result, the mean of the conditional distribution, $E[\mathbf{z}_{f_j}|\mathbf{r}_{p_k}]$, is equal to $\Lambda_{\mathbf{z}_{f_j}\mathbf{r}_{p_k}}\Lambda_{\mathbf{r}_{p_{kk}}}^{-1}\mathbf{r}_{p_k}$ (i.e., the linear least-square estimate of $\mathbf{z}_{f_j}$ using $\mathbf{r}_{p_k}$). The last step is to compute the expectation

$$E\left[\Lambda_{\mathbf{z}_{f_j}\mathbf{r}_{p_k}}\Lambda_{\mathbf{r}_{p_{kk}}}^{-1}\mathbf{r}_{p_k} \exp(\mathbf{r}_{p_k})\right] = \Lambda_{\mathbf{z}_{f_j}\mathbf{r}_{p_k}}\boldsymbol{\mu}_{\mathbf{y}_{p_k}} = \mathrm{Cov}(\mathbf{z}_{f_j}, \mathbf{y}_{p_k})$$

which, after rearranging terms, yields

$$\mathbf{\Lambda}_{\mathbf{z}_{f_m}\mathbf{r}_{p_n}} = \mathrm{Cov}(\mathbf{z}_{f_m}, \mathbf{y}_{p_n}) / \boldsymbol{\mu}_{\mathbf{y}_{p_n}}.$$

We note that the final equation is equivalent to a derivation provided by Buesing et al. [21] as their supplementary equation (6) to compute cross-covariances between Poisson observations and Gaussian *inputs*, instead of between joint Poisson and Gaussian *observations* (as was in our case). The remaining unimodal (i.e., Poisson-only) moment conversions that are required to compute $\mathbf{H}_{\mathbf{r}}$ are performed per equation (5) in section 2.2. For Bernoulli-Gaussian moment conversion equations, we refer the reader to section 3.3 equations (5) and (6) in Stone et al. [24].

### A.1.6 Generalized cross-term Hankel matrix with different horizons per observation

For ease of exposition, the derivation in section A.1.2 was provided for a cross-term Hankel matrix $\mathbf{H}_{\mathbf{zr}}$ that was formed with equal horizons for $\mathbf{z}$ and $\mathbf{r}$ as

$$\mathbf{H}_{\mathbf{zr}} := \mathrm{Cov}(\mathbf{z}_f, \mathbf{r}_p) = \begin{bmatrix} \mathbf{\Lambda}_{\mathbf{zr}_i} & \mathbf{\Lambda}_{\mathbf{zr}_{i-1}} & \cdots & \mathbf{\Lambda}_{\mathbf{zr}_1} \\ \mathbf{\Lambda}_{\mathbf{zr}_{i+1}} & \mathbf{\Lambda}_{\mathbf{zr}_i} & \cdots & \mathbf{\Lambda}_{\mathbf{zr}_2} \\ \vdots & \vdots & \cdots & \vdots \\ \mathbf{\Lambda}_{\mathbf{zr}_{2i-1}} & \mathbf{\Lambda}_{\mathbf{zr}_{2i-2}} & \cdots & \mathbf{\Lambda}_{\mathbf{zr}_i} \end{bmatrix}, \quad \mathbf{z}_f := \begin{bmatrix} \mathbf{z}_i \\ \vdots \\ \mathbf{z}_{2i-1} \end{bmatrix}, \mathbf{r}_p := \begin{bmatrix} \mathbf{r}_0 \\ \vdots \\ \mathbf{r}_{i-1} \end{bmatrix}.$$

In general, the rank of Hankel matrices formed from ideal data covariances can be shown to be the same as the state dimension associated with it [19, 20], i.e., $n_1 = \text{rank}(\boldsymbol{H}_{\mathbf{zr}})$ per equation (17) and $n_1 + n_2 = \text{rank}(\boldsymbol{H}_{\mathbf{r}})$ per equation (15). However, during system identification these Hankel matrices are formed from non-ideal empirical sample covariances and, as a result, are typically full rank. Nevertheless, we expect the singular values associated with real dynamics (e.g., the first $n_1$ singular values in $\boldsymbol{H}_{\mathbf{zr}}$) to be larger than subsequent singular values that are due to noise. Indeed, the goal of the SVD applied to Hankel matrices, e.g., in equations (15), (17), and (22), is to remove noisy singular values and only keep the largest singular values that are most likely due to real dynamics.

Given that the Hankel matrices formed during system identification are typically full rank, their rank is determined based on their dimensions, i.e., $\text{rank}(\boldsymbol{H}_{\mathbf{zr}}) = \min(i \times n_r, i \times n_z)$ and $\text{rank}(\boldsymbol{H}_{\mathbf{r}}) = i \times n_r$. Thus, the horizon parameter $i$ that is used to form the Hankel matrix plays an important role in its final dimensions, rank, and, consequently, on the maximum number of non-zero singular values that can be preserved after applying SVD. This, in turn, determines the maximum state dimension that can be learned for the resulting model. Thus, to provide more flexibility over the state dimensions that can be learned in each stage of PGLDM, we generalize the Hankel matrix $\boldsymbol{H}_{\mathbf{zr}}$ to support different horizon values for each of the observations, $i_z$ and $i_r$, such that

$$
\boldsymbol{H}_{\mathbf{zr}} = \begin{bmatrix} \boldsymbol{\Lambda}_{\mathbf{zr}_{i_z}} & \boldsymbol{\Lambda}_{\mathbf{zr}_{i_z-1}} & \cdots & \boldsymbol{\Lambda}_{\mathbf{zr}_{i_z-i_r+1}} \\ \boldsymbol{\Lambda}_{\mathbf{zr}_{i_z+1}} & \boldsymbol{\Lambda}_{\mathbf{zr}_{i_z}} & \cdots & \boldsymbol{\Lambda}_{\mathbf{zr}_{i_z-i_r+2}} \\ \vdots & \vdots & \cdots & \vdots \\ \boldsymbol{\Lambda}_{\mathbf{zr}_{2i_z-1}} & \boldsymbol{\Lambda}_{\mathbf{zr}_{2i_z-2}} & \cdots & \boldsymbol{\Lambda}_{\mathbf{zr}_{2i_z-i_r}} \end{bmatrix} \quad \text{with} \quad \mathbf{z}_f := \begin{bmatrix} \mathbf{z}_{i_z} \\ \vdots \\ \mathbf{z}_{2i_z-1} \end{bmatrix}, \mathbf{r}_p := \begin{bmatrix} \mathbf{r}_0 \\ \vdots \\ \mathbf{r}_{i_r-1} \end{bmatrix}.
$$

The observation horizon $i_r$ is also used when forming the Hankel matrix $\boldsymbol{H}_{\mathbf{r}}$, per equation (14). The additional flexibility gained from having different horizon values can be especially critical in scenarios wherein the dimensionalities of $\mathbf{z}$ and $\mathbf{r}$ are very different, such as in the case of our NHP analysis in section 4.2, where $n_z = 4$ and $n_r = 15$. We select the final horizons $i_z$ and $i_r$ via an inner cross-validation based on which values achieve the best decoding accuracy in the training data.

## A.2 Assumptions and generalizability of the block structure formulation

Here we explain the assumptions underlying the proposed block formulation of $\boldsymbol{A}$, as defined in equation (7), and discuss the model's generalizability. First, we assume that the private latents of $\mathbf{y}$ (i.e., $\mathbf{x}^{(2)}$) and of $\mathbf{t}$ (i.e., $\mathbf{x}^{(3)}$) do not drive the shared latents (i.e., $\mathbf{x}^{(1)}$) so as not to leak private dynamics into shared dynamics. Thus, we take $\boldsymbol{A}_{12} = \boldsymbol{0}$ and $\boldsymbol{A}_{13} = \boldsymbol{0}$. Second, we assume that the private dynamical latents of the two signals (i.e., $\mathbf{x}^{(2)}$ and $\mathbf{x}^{(3)}$) do not drive each other, and so we take $\boldsymbol{A}_{23} = \boldsymbol{0}$ and $\boldsymbol{A}_{32} = \boldsymbol{0}$. Finally, the asymmetrical decision to set $\boldsymbol{A}_{21} \neq \boldsymbol{0}$ and $\boldsymbol{A}_{31} = \boldsymbol{0}$ was dictated by the asymmetrical roles played by the primary (i.e., $\mathbf{y}$) and secondary (i.e., $\mathbf{t}$) time-series in our formulation. We initially made the assumption that $\boldsymbol{A}_{31} = \boldsymbol{0}$ to simplify the derivation of our method because it allowed the optional third stage (i.e., extraction of the dynamics private to $\mathbf{t}$) to operate independently of the first stage (i.e., extraction of the shared dynamics). However, this assumption was still in line with our notion of shared and private dynamics; if $\boldsymbol{A}_{31} \neq \boldsymbol{0}$, then the primary time-series ($\mathbf{y}$) would be informative (i.e., predictive) of the private dynamics in the secondary time-series ($\mathbf{t}$) through the influence of the shared dynamics $\mathbf{x}^{(1)}$ on future $\mathbf{x}^{(3)}$.

Although $\boldsymbol{A}_{21}$ fundamentally has a similar information leakage impact in the reverse direction, we chose to keep the more general form of $\boldsymbol{A}_{21} \neq \boldsymbol{0}$. Unlike the $\boldsymbol{A}_{31}$ case, a non-zero $\boldsymbol{A}_{21}$ does not couple the multi-stage learning. This is because we designate a primary time-series in our formulation (taken as $\mathbf{y}$ here) that acts as a predictor for the secondary time-series (taken as $\mathbf{t}$ here). Given this designation, we exclusively calculate all Hankel matrices using the primary/predictor time-series as the past observations. Specifically, we only compute $\boldsymbol{\Lambda}_{\mathbf{zr}_\tau} = \text{Cov}(\mathbf{z}_{k+\tau}, \mathbf{r}_k)$ (equation (8)) and not $\boldsymbol{\Lambda}_{\mathbf{rz}_\tau} = \text{Cov}(\mathbf{r}_{k+\tau}, \mathbf{z}_k)$. From a derivation perspective, this means that $\boldsymbol{A}_{21}$ only appears in stage 2 and not in stage 1, thereby never coupling the learning of private and shared dynamics. However, we clarify that if the user requires a strictly symmetrical definition of $\boldsymbol{A}$ wherein all latent states are fully decoupled from each other (i.e., a block-diagonal structure), our algorithm can, with a minimal modification to stage 2, cover the case where $\boldsymbol{A}_{21} = \boldsymbol{0}$. To learn a block diagonal $\boldsymbol{A}$, one would only need to compute the solution to $\underline{\boldsymbol{\Delta}}^{(2)} = \boldsymbol{A}_{22}\overline{\boldsymbol{\Delta}}^{(2)}$ instead of the original formulation $\underline{\boldsymbol{\Delta}}^{(2)} = [\boldsymbol{A}_{21} \quad \boldsymbol{A}_{22}]\overline{\boldsymbol{\Delta}}^{(1,2)}$ (see section 3.2.2 and equation (24)).

Finally, taking $\boldsymbol{A}_{21} \neq \boldsymbol{0}$ also has particular significance in the case when there are no private latents associated with $\mathbf{t}$, that is when $n_3 = 0$. In this case, taking $\boldsymbol{A}_{21}$ non-zero allows the block-structured state-space form defined in equations (6) and (7) to be assumed without any loss of generality. Specifically, when $n_3 = 0$, the latent states in our model describe the primary time-series $\mathbf{y}$ with a subset also explaining the secondary time-series $\mathbf{t}$. Formally, we define the true dimensionality of the shared states (denoted by $n_1$) based on the rank of the observability matrix for the pair $(\boldsymbol{A}, \boldsymbol{C}_{\mathbf{z}})$. It can be shown using linear systems theory that an invertible linear transformation of the latent states (i.e., a similarity transformation) always exists that can place the $n_1$ dimensional latent subspace that is observable via $\mathbf{t}$ as the first few dimensions of the latent space, thus giving the block-structured formulation of equation (7). This can be seen by applying Theorem 3.8 from *Subspace Methods for System Identification* [20] to the first two lines of equation (6). As a result, the blocked formulation of equation (7) is equivalent to the formulation from (6) and we can aim to learn our model in the form of equation (7) without any loss of generality. Thus, our choice to keep $\boldsymbol{A}_{21} \neq \boldsymbol{0}$ was also in part motivated by a desire to maintain a more general state-space form for the case when $n_3 = 0$.

### A.3 Noise statistics

Standard SSID algorithms (e.g., section 2.1) learn linear state-space models of the following form

$$
\begin{cases}
\mathbf{x}_{k+1} &= \boldsymbol{A}\mathbf{x}_k + \mathbf{w}_k \\
\mathbf{r}_k &= \boldsymbol{C}_{\mathbf{r}}\mathbf{x}_k + \mathbf{v}_k
\end{cases} , \tag{29}
$$

where state noise, $\mathbf{w}_k$, and observation noise, $\mathbf{v}_k$, are typically additive Gaussian noise and may have a non-zero instantaneous cross-covariance $\boldsymbol{S} = \mathrm{Cov}(\mathbf{w}_k, \mathbf{v}_k)$. SSID in general does not assume any restrictions on the noise statistics. However, the Poisson observation model (equations (4) and (6)) has no additive Gaussian noise for $\mathbf{r}_k$ and instead exhibits Poisson noise in $\mathbf{y}_k$ when conditioned on $\mathbf{r}_k$. This means that $\mathbf{v}_k = \boldsymbol{0}$ in equation (6), and thus $\boldsymbol{R} = \boldsymbol{0}$ and $\boldsymbol{S} = \boldsymbol{0}$. Imposing these constraints is important for accurate parameter identification for Poisson observations, but was not previously addressed by Buesing et al. [21]. Thus, we require our algorithm to find a complete parameter set $\Theta'$ that is close to the learned $(\boldsymbol{A}, \boldsymbol{C}_{\mathbf{r}}, \boldsymbol{C}_{\mathbf{z}}, \boldsymbol{b})$ by PGLDM *and* imposes the noise statistic constraints $\boldsymbol{R} = \boldsymbol{0}$ and $\boldsymbol{S} = \boldsymbol{0}$. To do this, inspired by Ahmadipour et al. [22], we form and solve the following convex optimization problem to satisfy the noise statistics requirements

$$
\begin{aligned}
\underset{\boldsymbol{\Lambda}_{\mathbf{x}}}{\text{minimize}} \quad & \|\boldsymbol{S}(\boldsymbol{\Lambda}_{\mathbf{x}})\|_F^2 + \|\boldsymbol{R}(\boldsymbol{\Lambda}_{\mathbf{x}})\|_F^2 \\
\text{such that } & \boldsymbol{\Lambda}_{\mathbf{x}} \succeq 0, \ \boldsymbol{Q}(\boldsymbol{\Lambda}_{\mathbf{x}}) \succeq 0, \ \boldsymbol{R}(\boldsymbol{\Lambda}_{\mathbf{x}}) \succeq 0
\end{aligned} \tag{30}
$$

where $\boldsymbol{\Lambda}_{\mathbf{x}} := \mathrm{Cov}(\mathbf{x}_k, \mathbf{x}_k)$ denotes the latent state covariance. Further, we enforce the following covariance relationships, derived from equation (1) [19], as constraints

$$
\begin{cases}
\boldsymbol{Q}(\boldsymbol{\Lambda}_{\mathbf{x}}) &= \boldsymbol{\Lambda}_{\mathbf{x}} &- \boldsymbol{A}\boldsymbol{\Lambda}_{\mathbf{x}}\boldsymbol{A}^T \\
\boldsymbol{R}(\boldsymbol{\Lambda}_{\mathbf{x}}) &= \boldsymbol{\Lambda}_{\mathbf{r}_0} &- \boldsymbol{C}_{\mathbf{r}}\boldsymbol{\Lambda}_{\mathbf{x}}\boldsymbol{C}_{\mathbf{r}}^T \\
\boldsymbol{S}(\boldsymbol{\Lambda}_{\mathbf{x}}) &= \boldsymbol{G} &- \boldsymbol{A}\boldsymbol{\Lambda}_{\mathbf{x}}\boldsymbol{C}_{\mathbf{r}}^T
\end{cases} . \tag{31}
$$

This approach has multiple benefits. First, it finds noise statistics that are consistent with the assumptions of the model (e.g., $\boldsymbol{R} = \boldsymbol{0}$). Second, it enforces the validity of learned parameters, i.e., parameters corresponding to a valid positive semidefinite covariance sequence (see section 4.4). It also enables state prediction (see appendix A.8.2). Combining the previously found parameters and the matrix $\boldsymbol{Q}$ that corresponds to the minimizing solution $\boldsymbol{\Lambda}_{\mathbf{x}}$ of equation (30), we have the full parameter set $\Theta' = (\boldsymbol{A}, \boldsymbol{C}_{\mathbf{r}}, \boldsymbol{C}_{\mathbf{z}}, \boldsymbol{b}, \boldsymbol{Q})$. We used Python's CVXPY package to solve the semidefinite programming problem defined in equation (30) [39, 40]. For all of our comparisons against PLDSID, we learned the noise statistics associated with the method's identified parameters using this approach, but keeping the rest of the algorithm the same.

### A.4 GLDM parameter equivalencies

Here we briefly discuss the equivalence of learnable parameters by SSID for linear dynamical models of the form in equation (1). In section 2.1 we stated that standard covariance-based SSID learned the parameters $\Theta = (\boldsymbol{A}, \boldsymbol{C}_{\mathbf{r}}, \boldsymbol{G}, \boldsymbol{\Lambda}_{\mathbf{r}_0})$, but not any of the noise statistics (e.g., covariances $\boldsymbol{Q}$ and $\boldsymbol{R}$). Indeed, there also exist SSID approaches that instead learn the following parameter set $(\boldsymbol{A}, \boldsymbol{C}_{\mathbf{r}}, \boldsymbol{Q}, \boldsymbol{R}, \boldsymbol{S})$ [19, 20]. Both sets of parameters are valid and knowledge of either completely

defines the model in equation (1). Moreover, the two sets of model parameters can be related to each other using equations (31) [19] – the same relationships we use as constraints in the convex optimization problem defined in appendix A.3 for learning the noise statistics of the PLDS model (4).

## A.5 Possibility of learning unstable modes in small data regimes

Subspace identification methods generally only converge to the correct system parameters asymptotically (see figure 4 in appendix A.11), as the empirically estimated covariances also converge to their true values [19]. For finite samples, however, there will always be some error in the learned parameters. Although such errors are generally benign, extreme scenarios can result in unstable state dynamics, i.e., the identified $A$ has at least one eigenvalue with magnitude larger than 1. We omitted any learned unstable models in our analyses, which was reflected in the reduced number of samples in the standard error of the mean (s.e.m). However, we rarely encountered unstable models; for example, in the results presented in figure 4, there were no unstable models for training set sizes typical of neuroscience datasets (i.e., 1e5 or 1e6 training samples).

## A.6 Possibility of accumulation of error in multi-staged learning

There are two aspects of the learning to consider. First, there is a signal-to-noise (SNR) consideration that is fundamental to learning methods in general. In stage 1, our method requires high SNR in the cross-correlations between future secondary time-series observations and past primary time-series observations. If most of the signal present in the primary time-series is not attributable to the shared latent states (i.e., the residual Hankel in equation (10) dominates), then most of $\boldsymbol{H_{zr}}$'s singular values will be small and stage 1 may have greater estimation error. If the reverse situation holds and the shared latent states explain most of the signal in the primary time-series, then the residual Hankel, $\boldsymbol{H}_r^{(2)}$ will mostly have small singular values, possibly resulting in estimation errors during stage 2. Inspection of the singular values prior to model parameter extraction can, however, help guide method usage. For example, if the singular values of $\boldsymbol{H_{zr}}$ are small, then this may indicate that the two time-series do not have shared dynamics and only stage 2 of the method is needed (i.e., standard GLDM). If the singular values of the residual Hankel are small, then this may indicate that almost all dynamics are shared and only stage 1 is needed.

A second consideration is a numerical one that could result in an accumulation of errors in downstream stages. If during stage 1 the modes are identified inaccurately because there is minimal shared dynamics between the two time-series and/or the training sample size is too small, then this estimation error could impact the computation of the residual matrix $\boldsymbol{H}_r^{(2)}$ thereby introducing error in stage 2. Examining the singular values of $\boldsymbol{H_{zr}}$ may, however, also help avoid this situation. Although the multi-stage learning can lead to error accumulation in some situations, it comes with the benefit that our method has the ability to more accurately identify the shared dynamics (when they exist) in stage 1, compared to existing GLDM methods. Moreover, Figures 1b, 2b, and 3b suggest that the impact of error accumulation is not severe and can be situation-dependent; for example, self-prediction performance of PGLDM reaches that of PLDSID in 1b, reaches very close to it in 2b, and exceeds it in 3b.

## A.7 Experimental details

Below we provide details on how simulations were generated and about our real data analyses.

### A.7.1 Simulations

For our synthetic data in section 4.1, we simulated generalized-linear observations from random models as per equation (6). For the simulations used to generate the results in Table 1, we fixed the number of shared and private latent states as $n_1 = 2, n_2 = 6$, and $n_3 = 4$. We randomly selected the observation dimensions with uniform probability from the following ranges: $10 \leq n_r \leq 15$ and either $5 \leq n_z \leq 10$, when the secondary observation was Gaussian, or $10 \leq n_z \leq 15$, when Poisson. For the simulations used to generate figure 2, we fixed the latent dimensions as $n_1 = 4, n_2 = 12$, and $n_3 = 4$ and randomly sampled observation dimensions with uniform probability from: $20 \leq n_r \leq 30$ and $5 \leq n_z \leq 10$. Across all simulations we used these dimensions to generate random model parameters $\Theta = (\boldsymbol{A}, \boldsymbol{C_r}, \boldsymbol{C_z}, \boldsymbol{b}, \boldsymbol{d}, \boldsymbol{Q})$. We constrained the complex eigenvalues (i.e.,

modes) of the state transition matrix $\mathbf{A}$ to have magnitudes uniformly distributed between $[0.93, 0.99]$ and phases uniformly distributed between $[0.019, 0.314]$. These restrictions correspond to stable, slow-decaying systems with time-constants within $[0.138, 0.995]$ seconds and frequencies within $[0.3, 5]$ Hz that are representative of various real time-series data, such as neural dynamics [41, 42].

For all Gaussian simulations we set the time-series mean (e.g., $\boldsymbol{b}$ or $\boldsymbol{d}$) to 0. Additive Gaussian noise covariances (e.g., $\boldsymbol{Q}$) were randomly generated to be positive definite matrices. For observation time-series, the observation matrix $C$ was generated to hit a target signal-to-noise, defined as the variance associated with latent states normalized by observation noise variance (e.g., $(\boldsymbol{C_z \Lambda_x C_z^T})/(\boldsymbol{\Lambda_\epsilon})$). Target SNR values were fixed at 10 for the simulations used in Table 1 or were randomly generated as $10^\alpha$ with $\alpha$ uniformly distributed between $[0, 2]$ for the simulations in figure 1. All Poisson simulations were generated on a 10 ms timescale with a baseline log rate (e.g., $\boldsymbol{b}$) randomly selected within $[0.5, 15]$ Hz. Observation matrix $C$ was scaled to achieve a desired per-dimension 95th percentile modulation depth of 10, where modulation depth is defined as $\exp(\boldsymbol{C}\mathbf{x}_k)$. All Bernoulli observations were also generated on a 10ms timescale with a baseline set to 0. In this scenario, the observation matrix $C$ was scaled such that the corresponding Gaussian latent defined by $\boldsymbol{C}\mathbf{x}_k$ would have a standard deviation approximately equal to 1.

To simulate a state noise covariance $\boldsymbol{Q}$ that adhered to the block-structure assumption defined in assumption 3.2, we leveraged the fact that latent $\mathbf{x}^{(3)}$ is completely decoupled from latents $\mathbf{x}^{(1)}$ and $\mathbf{x}^{(2)}$ in our model definition (as per equations (7) and (13), and assumptions 3.2 and 3.3). We chose to simulate the dynamics private to the secondary observation time-series as a separate 4-dimensional latent dynamical model. Specifically, we chose to generate one dynamical model corresponding to the latents $\mathbf{x}^{(1)}$ and $\mathbf{x}^{(2)}$, and a separate dynamical model corresponding to $\mathbf{x}^{(3)}$. This approach, however, only works for Gaussian secondary observations. For the Poisson-Poisson case we instead generated an unconstrained $\boldsymbol{Q}$. Despite this deviation from assumption 3.2, our method's ability to identify the shared dynamical modes was not impacted, as shown by the results in Table 1.

Finally, for all simulation experiments we use horizon values of 10 for both observations by default. For the simulations used in Table 1 we generated 25600 training samples, and for the simulations used in figure 1 we generated 2e6 samples, splitting 50/50 into training and test data.

### A.7.2   NHP Dataset 1: motor cortex recordings during reaching

All NHP analyses in figure 2 were performed on a public dataset released by the Sabes lab [17], using the following sessions from monkey I: 20160915/01, 20160916/01, 20160921/01, 20160927/04, 20160927/06, 20160930/02. We performed cross-validation using randomly-selected, non-overlapping subsets of 15 channels ($n_r = 15$) binned at 50ms resolution within each session. We used a nested inner cross-validation to select hyperparameters per fold based on the prediction CC of kinematics in the training data. Hyperparameters in this context were discrete horizon $i_\mathbf{r}$, continuous horizon $i_\mathbf{z}$, and time lag, which specifies how much the neural time-series should be lagged to time-align with the corresponding behavioral time-series [33, 43, 44]. We swept $i_\mathbf{r}$ values of 5 and 10 time bins, $i_\mathbf{z}$ values of 10, 20, 22, 25, 28, and 30 time bins; and lag values of 0, 2, 5, 8, and 10 time bins. We removed channels that had average firing rates less than 0.5 Hz or greater than 100 Hz. Similar to Lawlor et al. [45], we also removed channels that were correlated with other channels using a correlation coefficient threshold of 0.4.

### A.7.3   NHP Dataset 2: visual areas V1-V2 during stimulation

All NHP analyses in figure 3 were performed on a public dataset released by Zandvakili and Kohn [16, 18], using a randomly selected session 107l002p67. In the released dataset, each recording session consisted of repetitions of 1.28s of stimulus presentation (1 of 8 possible orientation gratings) followed by 1.5s of a blank screen. For our analysis, we only considered neural activity during periods of stimulation binned at 20ms resolution. Further, we performed five-fold cross-validated modeling within each stimulation condition. For both observation time-series, we removed channels that had average firing rates less than 0.5 Hz or greater than 100 Hz. Because there were more V1 units than V2 units, we performed cross-validation using randomly-selected, non-overlapping subsets of V1 units equal to the number of available V2 units, that is $n_\mathbf{r} = n_\mathbf{z} = |\text{V2 units}|$. We present averaged results across folds, stimulation conditions, and non-overlapping subsets, which we collectively refer to as "trials" for a total of 120 trials. No hyperparameter optimization was performed for this analysis; discrete horizons for SSID-based methods were set to a fixed value of $i_\mathbf{r} = i_\mathbf{z} = 5$.

## A.8    Model evaluation

We evaluated learning using two different metrics: (1) shared dynamical mode identification for models with the true shared latent state dimensionality and (2) predictive power of inferred latent states.

### A.8.1    Shared dynamical mode identification accuracy

To evaluate the learning of shared dynamics, we computed the normalized eigenvalue error as $\|\Psi_{\text{true}} - \Psi_{\text{id}}\|_2 \, / \, \|\Psi_{\text{true}}\|_2$, where $\Psi_{\text{true}}$ and $\Psi_{\text{id}}$ denote vectors containing the true (i.e., the eigenvalues of $\boldsymbol{A}_{11}$) and learned shared eigenvalues, respectively, and $\|\cdot\|_2$ denotes the Euclidean L2-norm. To compute the first metric for our baselines that did not explicitly model shared dynamics, we needed to select the $n_1$ modes identified from the primary time-series *only* that were the most representative of the secondary time-series. To do so, we first trained our baselines on the primary time-series observations and extracted the latent states. Then, we sorted these learned latent states based on their accuracy in predicting the secondary observations (appendix A.8.2). We computed the eigenvalues associated with the top $n_1$ most predictive latent states, which we considered as the shared modes identified by our baselines. This was only necessary for configurations where the learned latent dimensionality was greater than the shared dimensionality (i.e., $n_1 + n_2 > n_1$). For configurations wherein learned $n_x$ is smaller than true $n_1$, we substituted missing modes with 0 prior to computing the normalized error.

### A.8.2    Predictive power of inferred latent states

We also evaluate the predictive power of our learned models in two scenarios: (1) prediction of the secondary observation time-series from the primary time-series, a common use-case in neuroscience [2, 7, 46], and (2) one-step ahead self-prediction of the primary time-series from itself. Using models learned from the training data, we constructed recursive Bayesian filters to estimate the latent states in a test dataset. Because our analyses in figures 1-3 used Poisson observations as the primary time-series, we chose to use a Poisson point-process filter (PPF) [47] for state estimation. (We note that using the PPF for state estimation is only possible if the learned noise statistics are valid, see appendix A.3.) We denote the one-step ahead latent state prediction of $\mathbf{x}_k$ using all samples of $\mathbf{y}_k$ up to time $k-1$ by $\hat{\mathbf{x}}_{k|k-1}$. These state estimates can be used to predict both sets of (latent) observations as either $\boldsymbol{C}_{\mathbf{z}}\hat{\mathbf{x}}_{k|k-1}$ or $\boldsymbol{C}_{\mathbf{r}}\hat{\mathbf{x}}_{k|k-1}$.

In order to compare the predictive power of our learned models with those of our baselines (PLDSID and Laplace-EM), it was necessary to learn an observation model (e.g., $\boldsymbol{C}_{\mathbf{z}}$) for the secondary time-series. To do so, we first estimate the latent states in the training data using a PPF and then fit a regression model (scikit-learn; statsmodels) from the latent states to the secondary observation [48, 49]. For example, when the secondary observation, $\mathbf{z}_k$, is a continuous Gaussian time-series, the parameter $\boldsymbol{C}_{\mathbf{z}}$ was learned using ordinary least-squares such that $\boldsymbol{C}_{\mathbf{z}} = \mathbf{Z}\hat{\mathbf{X}}^T(\hat{\mathbf{X}}\hat{\mathbf{X}}^T)^{\dagger}$, where columns of $\mathbf{Z}$ and $\hat{\mathbf{X}}$ contain $\mathbf{z}_k$ and $\hat{\mathbf{x}}_{k|k-1}$ for all training timepoints $k$. To make all methods more comparable, we use the same approach to refit the secondary observation's model learned by PGLDM.

For continuous Gaussian observations, we quantify the decoding performance using correlation coefficient (CC). For discrete Poisson observations we instead evaluate prediction using the area under the receiver operating characteristic curve (AUC). Since all Poisson predictions were made using the recursive Bayesian filter's estimates of the latent states, our goal was to validate if our model could accurately predict the occurrence of point process events (versus no events) in a given time step when using all past observations of the primary time-series $\mathbf{y}_k$. For example, in the results shown in figure 2b-c we computed the probability of an event for the $m$-th dimension of $\mathbf{y}$ at time $k$ conditioned on all observations $\mathbf{y}_{1:k-1}$, as

$$
\begin{aligned}
P(\mathbf{y}_k^m > 0|\mathbf{y}_{1:k-1}) \quad &= \sum_{\mathbf{x}_k} p(\mathbf{y}_k^m > 0 \mid \mathbf{y}_{1:k-1}, \mathbf{x}_k) p(\mathbf{x}_k \mid \mathbf{y}_{1:k-1}) \\[2mm]
&\stackrel{(a)}{=} E_{\mathbf{x}_k \mid \mathbf{y}_{1:k-1}} \left[ p(\mathbf{y}_k^m > 0 \mid \mathbf{x}_k) \right] \stackrel{(b)}{=} E_{\mathbf{x}_k \mid \mathbf{y}_{1:k-1}} \left[ 1 - \exp(\exp(\mathbf{r}_k^m)) \mid \mathbf{x}_k \right] \\[2mm]
&\stackrel{(c)}{\approx} E_{\mathbf{x}_k \mid \mathbf{y}_{1:k-1}} \left[ \exp(\mathbf{r}_k^m) \mid \mathbf{x}_k \right] \stackrel{(d)}{=} \exp\left( \hat{\mathbf{r}}_{k|k-1}^m + \tfrac{1}{2}\boldsymbol{\Lambda}_{\hat{\mathbf{r}}_{mm}} \right)
\end{aligned}
$$

Table 2: Average learning runtime over 25 fits with 30484 training samples

| Method Name | Runtime in seconds ($\pm$ s.e.m) |
|---|---|
| PGLDM, $n_1 = n_x = 8$ (SSID / optimization) | $0.269 \pm 0.008$ / $0.080 \pm 0.005$ |
| PGLDM, $n_1 = 4, n_x = 8$ (SSID / optimization) | $0.253 \pm 0.005$ / $0.125 \pm 0.046$ |
| PLDSID, $n_x = 8$ (SSID / optimization) | $0.199 \pm 0.002$ / $0.063 \pm 0.011$ |
| Laplace-EM, $n_x = 8$ (100 iterations) | $109.656 \pm 0.662$ |
| Laplace-EM, $n_x = 8$ (1 iteration) | $1.097 \pm 0.007$ |

where in (a) we simplify using $\mathbf{y}_k$'s conditional independence from the past $\mathbf{y}_{1:k-1}$, in (b) we simplify based on $\mathbf{y}_k \mid \mathbf{x}_k \sim \text{Poisson}(\exp(\mathbf{r}_k))$, in (c) we use the Taylor series approximation of $\exp(\exp(\mathbf{r}_k))$ for small $\exp(\mathbf{r}_k)$, and (d) is simply the mean of a log-normal random variable. Note that $\hat{\mathbf{r}}_k = \boldsymbol{C}_{\mathbf{r}} \hat{\mathbf{x}}_{k|k-1} + \boldsymbol{b}$ and $\boldsymbol{\Lambda}_{\hat{\mathbf{r}}} = \boldsymbol{C}_{\mathbf{r}} \boldsymbol{\Lambda}_{\hat{\mathbf{x}}_{k|k-1}} \boldsymbol{C}_{\mathbf{r}}^T$, where $\hat{\mathbf{x}}_{k|k-1}$ is the current estimate for the state and $\boldsymbol{\Lambda}_{\hat{\mathbf{x}}_{k|k-1}}$ the estimate for the state-prediction covariance. We can similarly compute AUC for the prediction of secondary Poisson observations, as in figure 3.

### A.9    Stage 3 simulation results

We validated our algorithm's third stage for learning the dynamics private to the secondary observation $\mathbf{z}$ using simulations. We simulated 10 random models with the following dimensions $n_1 = n_2 = n_3 = 4, n_r = 20, n_z = 4$, with the primary/secondary observation pair being Poisson/Gaussian. The Gaussian observations were set to have SNR of 10 (see appendix A.7.1). We constrained the complex eigenvalues of the state transition matrix $\boldsymbol{A}$ to have magnitudes uniformly distributed between $[0.93, 0.99]$ (as before) and phases uniformly distributed between $[0.019, \pi]$. Using a training set size of 1e6 samples, we used PGLDM with all three stages to learn all sets of shared and private modes. We then computed the normalized eignevalue error between the ground truth private modes (i.e., eigenvalues of $\boldsymbol{A}_{33}$) and the identified private modes. We found an average normalized identification error of 1.12% for learning the dynamics associated with the latent states $\mathbf{x}^{(3)}$. For comparison, this is approximately equivalent to -2.05 on the y-axis of figure **1**.

### A.10    Computation time details

We compared the computational runtime efficiency between our method PGLDM (using either stage 1 only or both stages 1 and 2), PLDSID, and Laplace-EM on one session of NHP data binned at 50ms resolution (section 4.2 and appendix A.7.2). We repeatedly trained on 25 distinct time-series datasets and computed an average runtime for learning across all fits. Each dataset consisted of a 6097-by-15 matrix (timesteps-by-features) of Poisson observations and a 6097-by-4 matrix of Gaussian observations. All methods learned latent models of dimension 8 (i.e., $n_x = 8$). When both stages of PGLDM were used, we fixed $n_1 = 4$ while keeping $n_x = 8$. Laplace-EM was run for 100 iterations (the default setting) and learning times are reported both as an average for 100 iterations as well as for a single iteration. For PGLDM and PLDSID we separately report average running times for the SSID portion of the algorithm and for the convex optimization problem used to learn state noise, as outlined in appendix A.3. The results are presented in Table 2.

Most of the computational cost of our algorithm is involved in the matrix operations associated with 1) computing the necessary covariance/Hankel matrices, 2) performing the moment conversion, and 3) performing the SVD of the future-past Hankel matrices. To perform the moment conversion our method requires a covariance matrix for stacked future-past Poisson-Poisson observations (section 2.2) and a future-past Gaussian-Poisson Hankel matrix (section 3.2.1). Both of these empirical estimates of second-order covariances are computed using matrix multiplications which scale with the number of samples. As an example, we can consider the setup used for the computational cost analysis in Table 2, wherein $n_r = 15$ and $i_r = 10$ (horizon). The computed square Poisson-Poisson covariance matrix was of dimension $2 * n_r * i_r = 2 * 15 * 10$ and was the result of a matrix multiplication between two matrices of dimension (2*10*15)-by-6078, where $6078 = \text{timesteps} - 2 * i_r + 1$. Thus, this operation would scale linearly with the length of the training data. Similarly, the computational cost of this matrix multiplication scales linearly with feature dimension and horizon. The remaining operations (i.e., the SVD and the moment conversion itself) are functions of the latent-state dimension and the feature dimensions for each observation timeseries.

All running time analyses were performed on a 2020 Macbook Pro using 2 GHz Quad-Core Intel Core i5 CPU with 16GB of 3733 MHz RAM.

### A.11  Data Efficiency

We also investigated the data efficiency of our method by generating 20 random Poisson/Gaussian systems and studying the effect of training set size on learning. The random systems used here were generated using the same procedure as the systems used in figure 1 except for the latent dimensionalities of the primary signal. We instead uniformly sampled the total latent dimensionality of $\mathbf{r}$ as $1 \leq n_1 + n_2 \leq 10$ and the shared dimensionality as $1 \leq n_1 \leq n_1 + n_2$; we kept a fixed size for the secondary observations private states $n_3 = 4$. We used 1e3, 1e4, 1e5, or 1e6 samples to train models, and then tested them on 1e6 samples of independent held-out data (figure 4). We found that with increasing training set sizes the performance of our method improved, both in identification of shared dynamics and in overall predictive power. This is expected because empirically estimated covariances converge towards their true values with increasing training set sizes, thereby improving the overall performance of our algorithm.

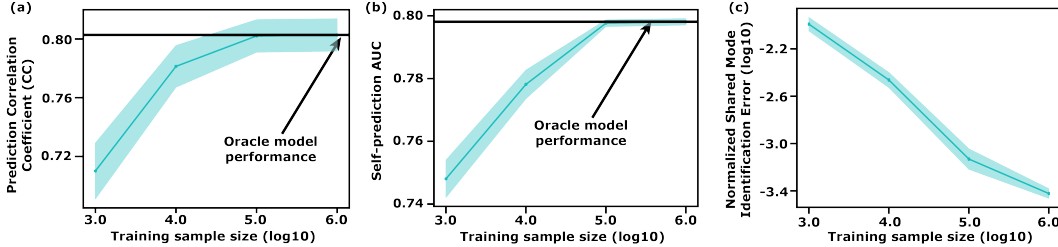

Figure 4: **PGLDM's overall performance improves with increasing number of samples, hitting peak performance with training set sizes of 1e5 samples.**. Solid traces show the mean and the shaded areas denote the standard error of the mean (s.e.m.) for each condition. (a-b) Predictive power of learned models as a function of training set size, with (a) depicting Gaussian observation decoding CC and (b) Poisson self-prediction AUC. (c) The normalized identification error of the shared dynamical modes (in log10 scale) as a function of training size.

### A.12  V1-V2 Results (time-series designation swap)

For completion we also present the complementary analysis to the results in figure 3, wherein we take neural population spiking activity from V1 as $\mathbf{y}_k$ and activity from V2 as $\mathbf{t}_k$. The rest of the analysis is kept the same as described in sections 4.3 and A.7.3, and the results are presented in figure 5. In this scenario, PGLDM does not make a substantial difference in predicting V2 activity from V1 activity when compared with PLDSID and Laplace-EM. Prior work using this same dataset has shown that the feedback direction, that is the V2 (past) to V1 (future) direction, generally exhibited higher correlations between the two time-series as compared to the feedforward direction [50]. We hypothesize that this finding might also explain the differences that we see in our results (i.e., the differences between figures 3 and 5).

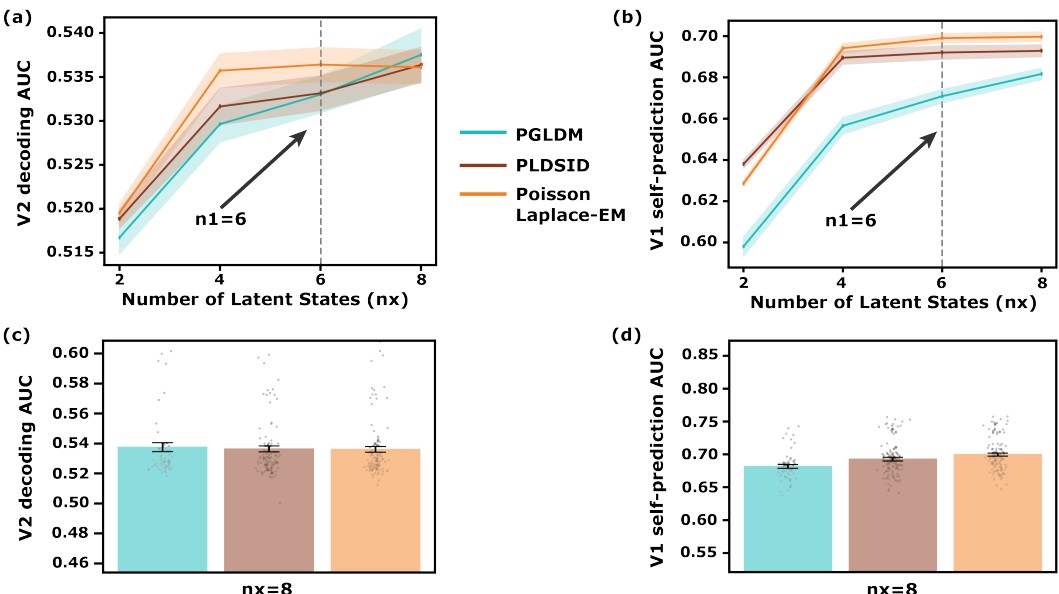

Figure 5: **All methods perform comparably in V2 decoding from V1 population spiking activity; PGLDM's V1 self-prediction performance improves with stage 2.** (a) Average cross-validated V2 decoding AUC (shaded areas denote s.e.m.) for models of different latent dimensions. (b) Same as (a) but for V1 one-step ahead self-prediction AUC. (c) V2 decoding AUC at $n_x = 8$. Whiskers correspond to s.e.m. Scatter points are individual trials. (d) Same as (c) but for V1 self-prediction.

