# OpenReview forum: "Spectral Learning of Shared Dynamics Between Generalized-Linear Processes"
_NeurIPS.cc/2024/Conference — NeurIPS 2024 poster_

### Official Review · Reviewer_95d5 · 2024-06-19

**Soundness:** 3
**Presentation:** 3
**Contribution:** 3
**Rating:** 5
**Confidence:** 3

**Summary:**

The work proposes a new linear dynamical system model that aims to model interaction between two time series by explicitly modeling their shared and private dynamics. Furthermore, authors incorporate generalized linear models in their observation model of the proposed dynamical system to handle different kinds of time series typically encountered in neuroscience (e.g., count-time series, real-valued time series etc.). Authors then extend standard covariance-based subspace system identification to learn the parameters of the proposed model and compared the performance of their proposed model with existing baselines on two experimental datasets and a simulation study.

**Strengths:**

- The paper is very well-written and the ideas are introduced in a natural progression making them easy to follow.
- Authors identify a gap in literature where existing methods do not model multiple time-series while explicitly accounting for shared and private dynamics when the observed time series are not real-valued.
- Authors propose an interesting dynamical system for modeling two different time series containing shared and private latents, whose parameters can be analytically estimated by extending existing covariance-based subspace system identification algorithms.
- The proofs provided in the paper are generally correct.
- Authors show improvement in the models' ability to predict one time series properties from the other time series when using their proposed dynamical system model which models the dynamics of the two time series jointly.

**Weaknesses:**

- The proof of the claim that the any general $A$, $C_r$, and $C_z$ can be decomposed into the block form structure described in equation (7) is incomplete. The proof provided in Appendix A.2 uses a well-known theorem, which states that for a non-fully observable linear dynamical system characterized by the system matrices $(A,C)$, there exists a non-singular linear transformation $T$ such that the new linearly transformed system having the new system matrices $A'=T^{-1}AT$ and $C'=CT$ have the following block structure:
$
TA^{-1}T = \begin{bmatrix} A_{11} & A_{12} \\\\ 0 & A_{22} \end{bmatrix}, CT = \begin{bmatrix}0& C_2\end{bmatrix}
$, where the pair $(A_{22}, C_2)$ are observable. Authors apply this theorem to the **special case** when the dimension of $x^{(3)}$ is zero, to reduce the matrices $A, C_z$ and $C_r$ to the desired block diagonal form, which is correct. But then, authors state that a similar argument works when the dimension of $x^{(3)}$ is non-zero. I am not sure how this proof of the special case of $x^{(3)}$ having zero-dimension can be extended to the general case of $x^{(3)}$ having non-zero dimension. The argument that the general matrices $A=\begin{bmatrix}A_{11}& A_{12}& A_{13}\\\\ A_{21}&A_{22}& A_{23}\\\\ A_{31}&A_{32}&A_{33} \end{bmatrix}$, $C_z = \begin{bmatrix} C_z^{(1)}&C_z^{(2)}&C_z^{(3)}\end{bmatrix}$, and $C_r=\begin{bmatrix}C_r^{(1)}&C_r^{(2)}&C_r^{(3)} \end{bmatrix}$ can always be reduced to the block-diagonal shown in equation (7) does not fall out as a trivial consequence of the theorem stated above, and requires an explicit proof. Naively applying the above theorem using the dimension of the observability matrix pair $(A, C_z)$ would show an existence of $T$ such that the new transformed system has the matrix with the following block structure: $A=\begin{bmatrix}A_{11}& A_{12}& A_{13}\\\\ A_{21}&A_{22}& A_{23}\\\\ 0&0&A_{33} \end{bmatrix}$, $C_z = \begin{bmatrix} 0&0&C_z^{(3)}\end{bmatrix}$.  I would ask authors to provide a rigorous mathematical proof showing that there always exist a non-invertible matrix $T$ such that a general $A, C_z$ and $C_r$ can be reduced to the block-diagonal form shown in equation (7). In the event, authors are unable to provide a proof, I would ask authors to modify the manuscript accordingly to reflect that the block diagonal structure is an assumption on the system model and remove the phrases "without loss of generality". In this review, I am going to continue on the assumption that the general matrices $A, C_r$ and $C_z$ cannot be reduced to the block-diagonal form (until shown otherwise by the authors through a rigorous mathematical proof) and the block-diagonal form is an assumption on the dynamical systems the authors are trying to estimate.
- The claims of the papers that the proposed model can be learnt for any generalized-linear process are overstated. To be able to learn the proposed models' parameters (described in equation (6)), the moment conversion equations described in section 3.2.3 are needed. Authors only provide the moment conversion equations for Gaussian, Poisson, and Bernoulli distributions. It is unclear how the model parameters can be learnt when, for example, $\mathcal{P}_{y|r}(y_k; g(r_k))$ is an exponential, gamma, or inverse-gaussian distribution. It may be the case that a general moment conversion equation may not exist for any arbitrary exponential distribution. Accordingly, I would ask the authors to town down their claims or provide a general moment conversion equation for any arbitrary exponential family distribution.

Minor

- Please do not use pythonic notation (without defining it first) for mathematical vector operations. For example in equation (11), R.H.S. of the equation is dividing a matrix by a vector. That is not a well-defined operation.  Similarly, equation (12) the operation $\mu_{t_{f_m}}\mu_{y_{p_n}}$ does not make sense, it should be $\mu_{t_{f_m}}\mu_{y_{p_n}}^T$. Please also state that $\ln(\cdot)$ in those operations is applied elementwise, as there exists well defined ways to define matrix functions of $\ln(\cdot)$ too.

**Questions:**

- The structure of the proposed $A$ matrix is a bit unintuitive and does not seem to satisfy intuitive properties one would expect from a dynamical system modeling shared and private latents. Intuitively, I would have expected the structure of the $A$-matrix to have the forms of these kinds: $A=\begin{bmatrix}A_{11}&0&0\\\\ 0&A_{22}&0 \\\\ 0&0&A_{33}\end{bmatrix}$, where the dynamics of the shared latent $x^{(1)}$  and the dynamics of the private latents $x^{2}$ and $x^{3}$ do not affect each other or the structure $A=\begin{bmatrix}A_{11}&A_{12}&A_{13}\\\\ A_{12}&A_{22}&0 \\\\ A_{31}&0&A_{33}\end{bmatrix}$, where the dynamics of the private latents $x^{3}$ and $x^{2}$ would only affect each other through the shared latent $x^{(1)}$. The structure proposed by the authors, i.e. $A=\begin{bmatrix}A_{11}&0&0\\\\ A_{12}&A_{22}&0\\\\ 0&0&A_{33}\end{bmatrix}$ has an inherent asymmetry in-built to its dynamics where the dynamics of the second private latent $x^{(2)}$ is affected by the shared latent state but the private latent $x^{(3)}$'s dynamics are only affected by itself. Why is this modeling choice justified? Why dynamics of only one of the private latent can be affected by the shared latent. Is there a domain-specific reason?
- Why does PGLDM always perform worse at self-prediction compared to other baselines (Figure 1 (b), 2(b), 3(b))?
- I am also curious to know that how the results look when decoding the activity V2 from V1, and V1 self-prediction using the system estimated by PGLDM in Section 4.3

**Limitations:**

Authors have provided a section discussing some limitations. A major limitation of this work that authors do not mention is the comparison with non-linear models which are also being used for decoding neural activity (such as neural data transformer by Ye & Pandarinath'21). I do not think a experimental comparison between these transformer architecture and the linear models proposed in this work is crucial, as the goal of these models have widely diverged with linear models being used more for understanding underlying neural dynamics, whereas the transform models being used for boosting decoding performance. Regardless, I think a mention of these non-linear methods should be made in the limitation sections along with the acknowledgement of a lack of comparison with these methods.

---

> ### Author Rebuttal · Authors · 2024-08-06
>
> We thank the reviewer for finding that the manuscript is well-written and addresses a gap in the literature. We also thank them for providing very comprehensive and helpful feedback. Below we reply to outstanding questions/concerns inline.
>
> > Generality claim on block form structure described in eq (7)
>
> We thank the reviewer for their feedback. The reviewer is exactly correct in all their points and we have revised the manuscript and Appendix A.2 to reflect these points. Briefly, as the reviewer points out, our proof on model generality provided in Appendix A.2 only applies to the case of $n_3=0$ (when learning the private dynamics of the secondary time-series is not of interest). When extending our method to the case of $n_3 > 0$ (i.e., optional stage 3), we add the following assumptions that are all motivated by having well-defined *private* dynamics but are not necessarily general:
> 1) We assume that the private dynamics of z ($x^{(3)}$) do not drive the shared dynamics of y and z ($x^{(1)}$), so as not to leak private z dynamics into shared dynamics. Thus we take $A_{13}=0$.
> 2) We assume that the private dynamics of the two signals ($x^{(2)}$ and $x^{(3)}$) do not drive each other, and for this reason we take $A_{23}=0$ and $A_{32}=0$.
> 3) Our assumption that $A_{31}=0$ was motivated by the definition of private dynamics in the context of our application. In our use case, the primary time-series ($y_k$) is used as the only observation during inference after the model is trained with both primary and secondary ($z_k$) time-series. In this setup, if $A_{31}$ was nonzero, the primary time series would be able to infer/predict, through $A_{31}$, private latents $x^{(3)}$ that are predictive of the private secondary time-series dynamics, undermining the notion of privacy. We do not encounter this situation with a non-zero $A_{21}$ and private primary dynamics $x^{(2)}$ because we do not use the secondary time-series as observation to estimate the primary time-series.
>
> We have revised Appendix A.2 and the main text to clarify these additional assumptions in the case of $n_3>0$, their motivation in terms of having well-defined “private” latents, and the fact that the complete generality in the block formulation of A is only for the $n_3=0$ case (stages 1 and 2). Future work interested in symmetrical inference applications may find ways to forgo some of the above assumptions.
>
> > Structure of the proposed $A$ matrix
>
> We thank the reviewer for their great question. We clarify a few points:
> - Our approach is also amenable to the reviewer’s first proposed structure $A=\begin{bmatrix}A_{11}&0&0\\\\0&A_{22}&0\\\\0&0&A_{33}\end{bmatrix}$. This is because in stage 2 we can extract $A_{22}$ directly by computing the least-squares solution $\underline{\Delta}^{(2)} \overline{\Delta}^{(2)\dagger}$ (section 3.2.2 and eq. (24)). However, we chose to keep $A_{21}$ to maintain the most general form (which in the case of $n_3=0$ is well-supported by theorem 3.8 [1], as discussed previously).
> - The second proposed structure $A=\begin{bmatrix}A_{11}&A_{12}&A_{13}\\\\A_{21}&A_{22}&0\\\\A_{31}&0&A_{33}\end{bmatrix}$ would not be suitable for our use case because unshared/private latents $x^{(2)}$ and $x^{(3)}$ would contribute to future timesteps of shared latents $x^{(1)}$ (via $A_{12}$ and $A_{13}$) – coupling shared and unshared; if we allow private dynamics to contribute to the future of shared dynamics, then this means they are not fully private. In our formulation we chose to keep the shared states completely isolated in their recurrent dynamics.
> - Arguably the most general form would have been to allow a nonzero $A_{31}$ in our formulation:  $A=\begin{bmatrix}A_{11}&0&0\\\\A_{21}&A_{22}&0\\\\A_{31}&0&A_{33}\end{bmatrix}$. However, as noted above, we chose to assume $A_{31}$ = 0 because otherwise the primary time-series would be informative/predictive of private dynamics in the secondary time-series, undermining the notion of privacy.
>
> > PGLDM self-prediction performance
>
> Thank you for the question. When using stage 1 only, our algorithm dedicates all the latent state dimensions ($n_1$) to the prioritized identification of shared dynamics. By doing so, the identified models are able to more accurately learn the shared dynamics and more accurately decode the secondary time-series. However, this means that self-prediction in stage 1 may suffer because the dominant dynamics in the primary time-series may be private rather than shared. After adding stage 2, however, PGLDM starts to learn the private dynamics and so its self-prediction performance improves in Figs 1b, 2b, and 3b, reaching very close to self-prediction of PLDSID once given enough latent state dimensions.
>
> > V2 decoding from V1; V1 self-prediction
>
> We thank the reviewer for their question. We have added a figure of the results in the PDF attached to our general response, which we will also include in the appendix. We originally performed our analysis only in the V2-V1 direction because prior work has shown that the feedback direction, that is the V2 (past) to V1 (future) direction, generally exhibited higher correlations between the two time-series as compared to the feedforward direction [2]. This might also explain the differences that we see in the results.
>
> > Additional comments
>
> We will include a discussion regarding nonlinear methods in the Limitations section as suggested. We have corrected the use of Pythonic notation indicated by the reviewer. We have also revised the text in lines 297-303 to tone down the claims about applicability to generalized-linear processes and now emphasize the need for a computationally tractable moment conversion equation. We thank the reviewer for the feedback.
>
> **References**:
>
> [1] Tohru Katayama. Subspace Methods for System Identification.
>
> [2] Semedo, J.D., Jasper, A.I., Zandvakili, A. et al. Feedforward and feedback interactions between visual cortical areas use different population activity patterns.

---

> ### Comment · Reviewer_95d5 · 2024-08-08
>
> I have read the rebuttal. I am still confused about the reasoning provided by the authors regarding the structure of A matrix. I am not entirely sure about the notion of private and shared dynamics authors are assuming. While I am okay with the authors' reasoning that $A_{12}=A_{13}=A_{23}=A_{32}=0$. The reasoning for $A_{31}=0$ seems very forced. I do not understand authors' reasoning that since they only use $y_k$ during inference, it is crucial that $y_k$ should not leak information about the other time-series $z_k$ through the shared dynamics and $A_{31}$ whereas since $t_k$ is not used during inference, it is okay for $t_k$ to leak information about the private dynamics through the shared dynamics. It seems very unlikely that neuronal dynamics in the brain would be affected by which time-series is being observed in the experiment. Either go all the way in assuming that shared dynamics and private dynamics do not leak information about each other or use the general structure. In my opinion, it seems more likely that $A_{31}=0$ is a simplifying assumption which is a *fine* assumption to make, as empirical evidence shows that the method work. I am keeping my original score.

---

> > ### Author Response · Authors · 2024-08-12
> >
> > We thank the reviewer for reading the rebuttal and for their thoughtful follow-up. We would like to clarify a few points:
> > 1. We acknowledge that assuming $A_{31}=0$ simplifies the derivation because it allows the optional third stage (i.e., extraction of the secondary time-series’ private dynamics) to operate independently of the first stage (i.e., extraction of the shared dynamics). We initially realized this relationship between $A_{31}$ and stage 3 while working on the derivation, and determined that this scenario (i.e., $A_{31} \neq 0$) would not be of interest in our use-case since having a non-zero $A_{31}$ also meant that y (or r) would be predictive of private t (or z) dynamics (i.e., $x^{(3)}$). We will, as per the reviewer’s point, more explicitly state in the manuscript that the $A_{31}=0$ model assumption simplifies the algorithm derivation.
> >
> > 2. We do not claim that $A_{31}$ and $A_{21}$ are fundamentally different in the sense of information leakage, but we clarify that their roles in our problem formulation are asymmetric for the following reason. In our formulation, y/r is designated as the primary time-series (predictor) and t/z as the secondary time-series (predicted). Given this designation, we always exclusively use y/r as the predictor and t/z as the predicted; this means that a non-zero $A_{21}$ does not allow t/z to directly predict future private dynamics of y/r (i.e., $x^{(2)}$). However, having a non-zero $A_{31}$ allows y/r to directly predict future private dynamics of t/z. This distinction between $A_{21}$ and $A_{31}$ arises because during learning we only ever compute the lagged cross-correlations $\Lambda_{zr_{\tau}} = \mathrm{Cov}(z_{k+\tau}, r_{k})$ (eq 8) and not $\Lambda_{rz_{\tau}} = \mathrm{Cov}(r_{k+\tau}, z_{k})$ -- as per the primary/secondary designation. Regardless, our solution can handle the symmetric case when both $A_{31}$ and $A_{21}$ are zero as well, see next item.
> >
> > 3. Regarding the $A_{21} = 0$ case, we agree with the reviewer that this is a very valid formulation as well and can be of interest to users. Our solution can learn a model with this block-diagonal $A$ structure with both $A_{21} = 0$ and $A_{31} = 0$ with a minor modification to the second stage, as we noted in our rebuttal. We have now also tested the block-diagonal $A$ case on a single session of the primate dataset in Fig 2 and confirmed that the block-diagonal $A$ formulation also identified the shared dynamics more accurately than baselines and performed comparably to the original non-block-diagonal $A$ formulation (eq 7). We will include discussion on the merits of this block-diagonal $A$ formulation in our manuscript, based on the reviewer’s excellent suggestion, so that users can choose whichever option they prefer in their application (a non-zero $A_{21}$ or $A_{21}=0$).

---

> > > ### Comment · Reviewer_95d5 · 2024-08-12
> > >
> > > I now better understand the authors' assumptions. I think the work will benefit by more explicitly discussing these assumptions. Regardless, these explanations do not alleviate my concerns over the bio-physical interpretation of these assumptions. One of the attractive property of linear dynamical models over non-linear models is their interpretability. The reasoning behind the assumption $A_{31}=0$ does not seem to stem from a bio-physical origin but as I understand (please clarify if I am wrong) from the fact that their is a primary time-series (which is designated by the "experimentalist/data scientist") and an additional "secondary" time-series which should not be able to predict the private dynamics ($x^{(2)}$) of the "primary" time-series. The designation of the "primary" and "secondary" time-series by itself sort of assumes a direction of causality/communication (I am using the words a bit vaguely) which in many cases might not be known a priori (many dynamical systems are indeed used for discovering this direction, see numerous works from the field of dynamic causal modeling). While this designation of time-series (e.g., from different brain regions) being primary and secondary might be perfectly reasonable for some applications (e.g., where there is a brain region sending information to another region without the other brain region providing feedback), there are also many cases where such an assumption might not be reasonable (e.g., in cases where brain areas communicate in both directions). This makes the model assumptions more restrictive. I still think the model assumption explored by the authors are interesting but can be restrictive.

---

> ### Author Response · Authors · 2024-08-13
>
> We thank the reviewer for following up. We are glad the assumptions are clearer now and that the reviewer finds them interesting. We will more explicitly discuss them in the manuscript, as suggested. The reviewer is correct that the reasoning behind the assumption $A_{31}=0$ is primarily due to designation of one time-series as primary and the other as secondary. Moreover, the reviewer raises an interesting point about the bio-physical interpretation of these assumptions. We agree with their interpretation that the primary/secondary designation implies, roughly, a direction of causality. The reviewer specifically highlights two scenarios: unidirectional interaction – with either known or unknown direction – and bidirectional interaction (e.g., feedforward and feedback). The designation of primary/secondary time-series has a bio-physical basis in applications wherein there is clear unidirectional interaction, consistent with the reviewer’s intuition. For example, one important use-case of our method is dissociating behaviorally-relevant neural dynamics. In this case, it can make sense to designate the neural activity as the primary time-series and behavior as the secondary time-series, with the reasonable bio-physical assumption that the brain drives behavior, in a unidirectional manner. Regarding the distinct use-case of modeling shared dynamics between brain regions, we agree with the reviewer that this designation makes most bio-physical sense in applications where the direction is from one brain region to the other, rather than being simultaneously in both directions (both feedforward and feedback). If there is a unidirectional interaction and the direction is known, the primary will become the upstream region activity and the secondary the downstream region activity. If the direction is unknown, one may be able to build models for each alternative designation and compare them according to the desired goodness-of-fit criterion. Finally, although the reviewer is correct that our current model formulation would not be able to handle bidirectional communication simultaneously, it can still be used to model each direction separately (i.e., model both possible designations of primary/secondary time-series separately). Regardless, we agree with the reviewer that being able to model both communication directions simultaneously could be even more general and will be an interesting direction for future investigation.
>
> Overall, we thank the reviewer for their very helpful feedback; we enjoyed these thoughtful discussions. We will definitely include explicit discussion regarding our model assumptions in the manuscript as well as their potential bio-physical implications – as the reviewer suggested.

---

### Official Review · Reviewer_m47S · 2024-07-12

**Soundness:** 3
**Presentation:** 4
**Contribution:** 3
**Rating:** 7
**Confidence:** 4

**Summary:**

In their study "Spectral Learning of Shared Dynamics Between Generalized-Linear Processes", the authors introduce a multi-stage spectral algorithm for learning the parameters of a model for two generalized-linear time-series with shared latent dynamics. Assuming the latent dynamics to contain both shared and private latent dimensions, the authors show that the shared dynamics can be identified from a decomposition of a Hankel matrix formed from the time-lagged covariances between the time series. Additional (second and third stage) analyses can be used to identify the latent spaces private to either of the two time-series. Thanks to moment-conversion schemes developed here and in previous work, their spectral method can be applied to time-series with Gaussian-, Bernoulli- and Poisson-distributed observations. On synthetic data and two datasets with primate neural recordings, the authors show that their method is able to accurately identify the private and shared dynamics. The method appears to perform particularly well on identifying the shared dynamics, leading to better predictions of one time series from the other than tested alternative methods. On the neural datasets, this allows better prediction of behavioural measurements and activity in other brain areas from spiking data.

**Strengths:**

The suggested method adds nicely both to the subspace identification literature and a growing body of literature in computational neuroscience that is interested in identifying shared subspaces between multivariate signals. The insight that in a linear dynamical system $(x,r,z)$ structured as in equation (7), the shared dynamics $A_{11}$ are identifiable from the time-lagged cross-covariances between $r$ and $z$ is really nice.

The core derivations of the suggested method appear solid and the experiments convincingly show that the method works (which I think for most parts is all it needs to do, as for generalized-linear time-series, to my knowledge there is little direct competition).

The manuscript is written both enjoyable and succinct. In particular the methods section -- including corresponding supplementary information -- is clear and well-written (minor comments on clarity of the results section below).

**Weaknesses:**

As the authors remark themselves, the research question addressed here is primarily interesting for neuroscientific applications, and may hence appear somewhat niche for a larger machine learning conference. This is not a strong reservation though, in particular given the roots of the NeurIPS conferences. The contribution to the subspace identification literature should make it worthwhile to a larger audience.

I am not sure if I can agree with the WLOG statement in Def 3.1 and section A.2 as it is currently written. The generality claim appears to rest on the possibility of $n_2 = n_3 = 0$ and hence $n_x = n_1$ - but in this case, what is the point of this work? I feel that the authors first should introduce the structure of shared and private latent spaces (which is not yet established by the preceding section 3.1) with the respective dimensionality split $n_x = n_1 + n_2 + n_3$. Then, given that assumption of the structured latent space with fixed $(n_1, n_2, n_3)$, equation (7) would be general.
The assumption A.1 is important for the algorithm and should be stated in the main text, ideally in definition 3.1 or in a small remark following it.

The multi-stage nature of the procedure feels somewhat prone to accumulation of errors -- since this is a spectral method this will hold particularly for small sample sizes. More specifically, I am a somewhat worried about numerical stability following the subtractions and subsequent matrix decompositions in equations (10) and (22) - any comments on this in the presence of estimation noise, in particular if one of the two terms shown in eq (19) to make up H dominates the other, e.g. in terms of Froebenius norm?

There is a typo in lines 822 (dropped a subscript '11' from one of the occurrences of the first block of A), and a missing subscript 'r' on matrix $C_r$ in line 967.

**Questions:**

I feel the first experiment (comparisons against Laplace-EM / PLDSID / bestLDS) could be a little better explained - are the comparisons trained only on the data of the primary time-series? The text only describes them as "models of the primary time-series as learned with either Laplace-EM or [...]". Is PSID trained on both primary and secondary time-series? A citation for Laplace-EM would be helpful. Also, what is the number of data points generated for the results in table 1?

I am a little surprised about the bad performance of EM in Fig 1 compared to PLDSID - commonly, one would initialize a Poisson-EM fit with the much cheaper PLDSID fit and get at least a mild improvement. Are the shown results related to the 'Laplace'-part of the Laplace-EM algorithm?

Could it be helpful to define $H$ as $H_r$ ? This notation would deviate from the classical literature only dealing with a single time series, but be consistent with the eventual definition of $H_z$ for step 3. It feels more natural to define $H$ as the Hankel matrix of the jointly-Gaussian stacked $[r, z]$, with $H_r$, $H_z$ and $H_{zr}$ collecting specific blocks.

Did the matrices $Q$ described in line 998 conform with assumption A.1 ?

In A.7.1, why describe the norm as Froebenius rather than Euclidean when it's about vectors?

Choosing red and green as important colors for the figures is not ideal for colorblind people.

**Limitations:**

The authors discuss limitations of their work in a dedicated section.

The authors state that by the fundamental nature of their work, they see no direct positive or negative societal consequences. I tend to disagree with the authors on this point, due to the possible use of works such as this one in Brain Computer Interfaces. Those hold promise to greatly benefit people with bodily limitations such as locked-in patients.

---

> ### Author Rebuttal · Authors · 2024-08-06
>
> We thank the reviewer for their close read of our manuscript and for finding it enjoyable with a well-written methods section. Below we address comments/questions inline.
>
> > Feedback on Def 3.1, section A.2, & assumption A.1
>
> We thank the reviewer for their suggestions. We clarify that the WLOG in Def 3.1 and section A.2 was to suggest that our algorithm could cover the learning of a GLDM for the single observation time-series as a special case (i.e., using stage 2 only with n1=n3=0, and n2=nx). We now realize this was unclear and have removed the WLOG statement in section 3.1 and revised section A.2 to make our intended meaning clearer. We will also move Assumption A.1 to section 3.1.
>
> > accumulation of errors in the presence of estimation noise
>
> We thank the reviewer for their great question. There are two aspects of the learning we wish to comment on.
>
> First there is a signal-to-noise (SNR) consideration that is fundamental to learning methods in general. In stage 1 our method requires high SNR in the cross-correlations between future secondary time-series observations and past primary time-series observations. If most of the signal present in the primary time-series is not attributable to the shared latent states (i.e., the residual Hankel in eq (19) dominates), then most of $H_{zr}$’s singular values will be small and stage 1 may have greater estimation error. If the reverse situation holds and the shared latent states explain most of the signal in the primary time-series, then the residual Hankel $H_{r}^{(2)}$ will mostly have small singular values, possibly resulting in estimation errors during stage 2. Inspection of the singular values prior to model parameter extraction can, however, help guide method usage. For example, if the singular values of $H_{zr}$ are small, then this may indicate that the two time-series don’t have shared dynamics and only stage 2 of the method is needed (i.e., standard GLDM). If the singular values of the residual Hankel are small, then this may indicate that almost all dynamics are shared and only stage 1 is needed.
>
> A second consideration is a numerical one that could, as the reviewer stated, result in an accumulation of errors in downstream stages. If during stage 1 the modes are identified inaccurately because there is minimal shared dynamics between the two time-series and/or the training sample size is too small, then this estimation error could impact the computation of the residual matrix $H_{r}^{(2)}$, introducing error in stage 2. Examining the singular values of $H_{zr}$ may, however, also help avoid this situation. Although the multi-stage learning can lead to error accumulation in some situations, it comes with the benefit that our method has the ability to more accurately identify the shared dynamics (when they exist) in stage 1, compared to existing GLDM methods. Finally, Figs 1b, 2b, and 3b suggest that the impact of error accumulation is not severe and can be situation-dependent; for example, self-prediction performance of PGLDM reaches that of PLDSID in 1b, reaches very close to it in 2b, and exceeds it in 3b. We will add a supplementary section discussing these points and add error accumulation to the Limitations section as a potential downside.
>
> > clarification on the first experiment
>
> Thank you for the feedback. We’ve revised the manuscript (sections 4.1 and A.6.1) to include more details regarding the first experiment. PSID and our method were trained on both the primary and secondary time-series because these algorithms can consider both time-series for training. All other baselines used only the primary time-series, as these methods only model a single data source. We’ve now included a citation in Table 1 to the GitHub library for the Laplace-EM algorithm that we used (previously cited only as a footnote). Lastly, we simulated 25600 timesteps for each configuration.
>
> > bad performance of EM in Fig 1
>
> We thank the reviewer for their great question. It is possible that the global Laplacian approximation underlying the EM algorithm in [1] may explain its performance in Fig 1. In this widely-used EM implementation [1], the global posterior (i.e., the probability of the latents $x$ conditioned on all observations $y_{1:T}$) is approximated as a Gaussian distribution so that it can be computed. It may be the case that the accuracy of the Gaussian approximation is poor for the posterior related to the Poisson observations in Fig 1, impacting overall model performance.
>
> > Structure of synthetic $Q$ matrices
>
> We thank the reviewer for the great question. For most of our simulations the $Q$ matrices did conform to assumption A.1. We deviated from the assumption only for the Poisson-Poisson simulations used in Table 1. However, as shown by the results in Table 1, this did not significantly impact our method's identification of shared dynamical modes. In most cases we simulated the block $Q$ structure by simulating the dynamics private to the secondary observation time-series as a separate latent dynamical model. Because latent $x_{3}$ is completely decoupled from $x_{1}$ and $x_{2}$ in our model definition (eqs (7) and (13), assumptions A.1 and A.2), we chose to generate one dynamical model corresponding to the latents $x_{1}$ and $x_{2}$ and a separate dynamical model corresponding to $x_{3}$. This approach generates a block $Q$ but only works for Gaussian observations, which is why the Poisson-Poisson case was different. We will add this clarification to section A.6.1.
>
> > Additional comments
>
> We have fixed the typos in lines 822 & 967, renamed $H$ to $H_r$, renamed the Frobenius norm to l2-norm, and will plan to adjust our plot colors in the final manuscript appropriately. We thank the reviewer for their recommendations. We also thank the reviewer for their assessment of our work and its applicability in the BCI domain. We will include this point in the manuscript as a potential benefit.
>
> **References**:
>
> [1] Linderman lab SSM library

---

> > ### Comment · Reviewer_m47S · 2024-08-13
> >
> > I thank the authors for their response and the clarifications. I will keep my rating as is.

---

### Official Review · Reviewer_3CeQ · 2024-07-12

**Soundness:** 3
**Presentation:** 3
**Contribution:** 3
**Rating:** 7
**Confidence:** 4

**Summary:**

In this paper, the authors propose spectral learning method for learning shared latent subspaces between multiple observations. The resulting model leverages the novel construction of Hankel matrix between observations from different processes. To promote discovery of shared subspaces, the authors propose the prioritised-extension of generliased linear dynamical modeling, which prioritises explaining the primary process using the shared subspace before learning the private subspaces specific to each process (again, priority is given to the primary process). The resulting model is demonstrated on a number of neural data benchmarks.

**Strengths:**

- The paper is clearly written, with clear pointer to mathematical details.
- Empirical studies is comprehensive and shows convincing results.

**Weaknesses:**

- Only spectral method has been proposed for learning in the proposed multi-observation GLDM. However, one unique strength of spectral methods is that the learned dynamics given spectral methods can be used as initial parameters for EM-based methods. It would be great to see if such extension is going to further improve the capability of the model.

**Questions:**

See Weaknesses.

**Limitations:**

See Weaknesses.

---

> ### Author Rebuttal · Authors · 2024-08-06
>
> We thank the reviewer for finding our manuscript clearly written with comprehensive and convincing experimental results. The reviewer raises a great question regarding initializing EM with our learned parameters. To the best of our knowledge, no existing EM algorithm can perform dissociation of shared vs. private dynamics and therefore there are no suitable EM algorithms that we could initialize with our parameters and use out-of-the-box.
>
> Specifically, there are two key features in our PGLDM learning approach: 1) we learn a model with the specific block structure defined in equation (7)  (i.e., $A_{11}$, $A_{21}$, $A_{22}$, and $A_{33}$) that allows shared and private dynamics to be dissociated within the model as distinct states $x^{(1)}$, $x^{(2)}$, and $x^{(3)}$, and 2) we learn this model in multiple consecutive stages such that the learning of shared dynamics can be prioritized in the first stage. Because existing EM methods do not allow for either the block-structure in eq. (7) or the prioritized learning of shared dynamics, we hypothesized that even if we initialize these methods with our parameters, they would not maintain the desired prioritized structure of the dynamical matrix $A$ (eqs. (6) and (7)).
>
> To test this hypothesis we used one of our simulations from Fig 1 to learn model parameters with our method at the true shared latent dimension (i.e., $n_x=n_1$) and initialize Laplace-EM (one of our baselines) with the learned $A$ and $C_r$ parameters. We ran EM for 50 iterations and computed the eigenvalue error of the final identified modes. We compared the resulting error with the error associated with the $A$ our method had identified and the error after running Laplace-EM with random initialization. We did this experiment 10 times with different folds of 40,000 training samples, training only on the primary observation time-series. Results are presented in the table below:
>
> |Method ($n_x=n_1$)|Normalized eigenvalue error mean &plusmn; STE (log10)|
> |-|-|
> |PGLDM (stage 1)|**-2.408 &plusmn; 0.0582**|
> |Lap-EM (random init.)|-0.9351 &plusmn; 0.0074|
> |Lap-EM (PGLDM init.)|-1.0476 &plusmn; 0.0333|
>
> We can see that initializing with our model parameters allowed EM to identify the modes more accurately than when randomly initialized, but it still had higher error than our method – as we hypothesized. There could be multiple reasons for this. Standard EM learns model parameters by maximizing the observation log-likelihood. For example, the final learned $A$ may deviate from our identified sublocks because it may be beneficial for EM to mainly learn private dynamics if they are more dominant than shared dynamics in the observation time-series. This is because standard EM has no way of prioritizing the learning of shared dynamics, i.e., our method’s second feature noted above. Also, even when EM is learning both shared and private dynamics, it will have no requirement/way to dissociate them into separate states because it lacks a block-structured approach (our method’s first feature noted above). So again EM can deviate from the clear dissociation that the sublocks identified by our method allows and instead learn states that reflect a mixture of both private and shared dynamics. Taken together, only an EM algorithm that can dissociate shared vs. private dynamics during optimization would be beneficial for being initialized with our PGLDM’s learned parameters, as also suggested by our new analysis. However, to the best of our knowledge, such an EM algorithm does not exist and the derivation/implementation of such an algorithm is better suited for future work (i.e., deriving an EM algorithm for the block structure proposed in eqs (6) and (7) or modifying the optimization cost function to prioritize shared dynamics by being multi-staged, for example).

---

### Official Review · Reviewer_Hfka · 2024-07-13

**Soundness:** 3
**Presentation:** 3
**Contribution:** 2
**Rating:** 5
**Confidence:** 3

**Summary:**

This paper introduces a novel multi-step analytical subspace identification algorithm for Generalized-Linear Dynamical Models (GLDMs) to model shared and private dynamics within two time-series data sources. The proposed algorithm effectively decouples shared and private dynamics, demonstrating superior performance in simulations compared to existing methods. The algorithm's efficacy is validated using synthetic data and two non-human primate neural datasets, showcasing improved decoding accuracy of one time-series from the other.

**Strengths:**

* The paper is well-written and has a smooth and concrete flow.
*  The paper presents a novel and effective approach to model shared and private dynamics with SSID theory.
* The experimental section is sufficient.

**Weaknesses:**

* The multi-step nature of the algorithm may introduce significant computational overhead, potentially limiting its scalability.
* The method assumes time-invariant dynamics, which may not hold for all neural data, potentially affecting its generalizability.
* While the algorithm is validated on neural datasets, its applicability to other domains or types of data is not explored, which could limit its broader impact.

**Questions:**

* I keep wondering what's the scientific meaning of modeling both the shared vs. private dynamics? Otherwise this would undermine the contribution of this whole work.
* How does the computational complexity of the proposed multi-step algorithm compare with existing GLDM learning algorithms, especially in terms of runtime and scalability?

**Limitations:**

To my knowledge and understanding, there is no potential negative societal impact of this work. Other limitations please see “Weaknesses”.

---

> ### Author Rebuttal · Authors · 2024-08-06
>
> We thank the reviewer for finding our paper well-written, our approach novel and effective, and the experimental section sufficient. Below we address outstanding questions and comments inline.
>
> > The method assumes time-invariant dynamics, which may not hold for all neural data, potentially affecting its generalizability.
>
> We thank the reviewer for raising this point and agree that the time-invariance assumption is a limitation. We included a discussion of this point in the Limitations section (line 277) and proposed two possible solutions: 1) intermittent model refitting, and 2) model adaptive extensions. For 1), one can refit the model after a predetermined duration of time, for example every day is typical in brain-computer interfaces. For 2), one can gradually update the model parameters by, for example, incorporating a learning rate parameter that weighs recent observations more heavily and gradually forgets past observations, similar to previous adaptive subspace methods [1]. We will revise the limitations section to include these discussions on possible extensions. We are also happy to include any other points the reviewer feels will be helpful.
>
> > I keep wondering what's the scientific meaning of modeling both the shared vs. private dynamics? Otherwise this would undermine the contribution of this whole work.
>
> We thank the reviewer for their excellent question. We provide our explanation within the context of neuroscience but the same reasoning can apply to analogous situations from other application domains.
>
> Modeling the shared dynamics is useful for studying how the brain encodes a particular behavioral/cognitive/affective process of interest, and for accurately decoding said process in brain-computer interfaces (BCIs). In general, the brain is simultaneously involved in several tasks or activities; for example, one may be moving their arm toward an object while also speaking and feeling excitement. As such, neural activity is a rich mixture model composed of multiple processes happening concurrently, thus necessitating the explicit dissociation of dynamics that are shared with a particular process of interest (e.g., movement). Another application, as reviewer m47S alluded to, is the development of BCIs. BCIs decode a behavioral/cognitive/affective process of interest from neural activity in real time. For example, BCIs can be used to restore lost motor function in paralysis, such as by decoding the patient’s movement intentions from neural activity to move a cursor on a computer screen. Dissociating and modeling the shared dynamics enables more accurate real-time decoding for BCIs as suggested by Fig. 2a, and is also more computationally efficient for real-time implementation due to the lower dimensionality required for the states, as suggested by Figs. 1a, 2a, and 3a.
>
> Modeling the private dynamics separately, as an additional stage, is helpful for studying neural activity in its entirety (not just the shared component), for example to build a generative model of neural activity. Beyond generative modeling, this is also important for basic science investigations on the role of private neural dynamics during processes of interest. For example, some private dynamics that are not directly representing movement kinematics (e.g., shared with kinematics measures such as position and velocity) may be involved in time-keeping or in higher level cognitive planning during a movement task.
>
> > [From Weaknesses] The multi-step nature of the algorithm may introduce significant computational overhead, potentially limiting its scalability. [From Questions] How does the computational complexity of the proposed multi-step algorithm compare with existing GLDM learning algorithms, especially in terms of runtime and scalability?
>
> We thank the reviewer for their great question. We have now expanded Table 2 to include computational time results for PLDSID and for PGLDM when using both stage 1 and stage 2 (vs. stage 1 only). To ensure consistency in hardware settings across all test conditions, we reran the entire experiment again and have updated the existing values in Table 2 to our latest results. We present the new table results here for convenience:
>
> |Method|Running time in seconds (mean &plusmn; STE)|
> |-|-|
> |PGLDM, $n_1=n_x=8$ (SSID / optimization)|0.269 &plusmn; 0.008 / 0.080 &plusmn; 0.005|
> |PGLDM, $n_1=4, n_x=8$ (SSID / optimization)|0.253 &plusmn; 0.005 / 0.125 &plusmn; 0.046|
> |PLDSID, $n_x=8$  (SSID / optimization)|**0.199 &plusmn; 0.002 / 0.063 &plusmn; 0.011**|
> |Laplace-EM, $n_x=8$ (100 iterations) |109.656 &plusmn; 0.662|
> |Laplace-EM, $n_x=8$ (1 iteration)|1.097 &plusmn; 0.007|
>
> Our method is more efficient than EM, which is iterative. Its run time is also only slightly higher than standard PLDSID (on the order of 10ths of a second). So in many practical applications the multi-stage nature should not pose a significant running time burden because most operations in each stage are analytical rather than numerical/iterative. Indeed, this computational efficiency compared with numerical or iterative approaches for fitting GLDMs, like EM, is a major advantage of subspace learning approaches such as PLDSID and our PGLDM. Finally, in terms of scalability within subspace learning approaches, PGLDM and PLDSID would scale similarly in terms of computational cost. We provide a more comprehensive explanation of this in supplementary section A.9, but briefly, we expect our method’s learning runtime to scale linearly as a function of training sample size, observation dimension, and horizon – similar to PLDSID.
>
> **References**:
>
> [1] Parima Ahmadipour, Yuxiao Yang, Edward F. Chang, and Maryam M. Shanechi. Adaptive tracking of human ECoG network dynamics.

---

### Author Rebuttal · Authors · 2024-08-06

We thank the reviewers for taking the time to review our submission and for providing helpful feedback, suggestions, and discussion regarding our work. We were encouraged to hear that reviewers found our manuscript “enjoyable” (reviewer m47S) and “well-written” (reviewers Hfka, m47S, and 95d5), with a “comprehensive” (reviewer 3CeQ) experimental section demonstrating that our “interesting” (reviewer 95d5) and “novel” (reviewer Hfka) approach works.

We provide responses to each reviewer’s comments and questions inline in each of our rebuttals below. We have also made revisions to the manuscript as needed to further address reviewer comments. Lastly, we performed a few new analyses for the manuscript to address some reviewer comments, and are including these as a PDF with a revised table 2 (with new running time analysis results), a revised figure 3 (additional barplots for V2-V1 analysis), and a new supplementary figure to complement figure 3 (V1-V2 analysis). Tables/figures are presented in that listed order. The color scheme in the attached figure, as well as all figures in the manuscript, will be updated in the final version to improve accessibility for color blind individuals, as advised by reviewer m47S.

---

### Decision · Program_Chairs · 2024-09-25

**Decision:**

Accept (poster)

**Comment:**

This paper describes a subspace ID method for characterizing generalized dynamical models.  All four reviewers felt that it was above the threshold for acceptance, and I'm pleased to report that it has been accepted to NeurIPS.  Congratulations!  Please revise the manuscript according to the reviewer comments and discussion points.